# The Southern Ocean Time Series: A climatological view of hydrography, biogeochemistry, phytoplankton community composition, and carbon export in the Subantarctic Zone

Elizabeth H. Shadwick[1,2], Cathryn A. Wynn-Edwards[1,2], Ruth S. Eriksen[1,2,3,4], Peter Jansen[3], Xiang Yang[2,4], Gemma Woodward[2], and Diana Davies[2]

[1]CSIRO Environment, Hobart, TAS, Australia
[2]Australian Antarctic Program Partnership, Hobart, TAS, Australia
[3]CSIRO National Collections and Marine Infrastructure, Hobart, TAS, Australia
[4]Institute for Marine and Antarctic Studies, University of Tasmania, Hobart, TAS, Australia

**Correspondence:** Elizabeth H. Shadwick (elizabeth.shadwick@csiro.au)

**Abstract.** The Southern Ocean Time Series (SOTS) provides highly temporally resolved observations of the physical, chemical and biological variability in the upper ocean, as well as the export of particulate carbon to the ocean interior, in the Subantarctic region south of Australia. The SOTS observatory focuses on the Subantarctic region because of its importance in the formation of mode water and the associated uptake and storage of anthropogenic heat and carbon. The region is also critical for the supply of oxygen to the interior and the export of nutrients to fuel primary production in broad areas of the low latitude global ocean. The SOTS observatory is the longest running multidisciplinary initiative in the open Southern Ocean, and has delivered high quality observations from the surface to the seafloor for more than a decade, and for some parameters, over two decades, using two deep-water moorings. The moorings are serviced annually, providing additional opportunities for shipboard sampling and sensor validation and calibration. Using observations collected at the SOTS site between 1997 and 2022, the seasonal variability in upper ocean hydrography, biogeochemistry, phytoplankton community composition, and biodiversity, along with carbon export to the ocean interior are presented. This climatological view of the region is complemented by a review of recent findings underpinned by observations collected by the SOTS observatory and highlighting the ongoing need for long time-series to better understand the Subantarctic ocean and its response to a changing climate.

## 1 Introduction

The Southern Ocean plays a predominant role in the movement of heat and carbon into the ocean interior, thereby moderating global average surface climate, variability, and rate of change. The oceanic uptake of excess anthropogenic heat and carbon dioxide ($CO_2$) drives changes in ocean ecosystems via warming (e.g. Couespel et al., 2021), stratification, acidification (e.g. Mortenson et al., 2021), and ventilation (e.g. Gnanadesikan et al., 2007), with an unknown mix of negative and positive consequences. The Southern Ocean Time Series (SOTS), a facility of the Australian Integrated Marine Observing System (IMOS) and a component of the global OceanSITES network, acquires data that allows these processes to be quantified in a region where they are most intense and relatively poorly understood. The oceanic uptake of heat and $CO_2$ varies over many

timescales from diel insolation cycles, to daily or weekly weather events in the atmosphere (e.g., Schulz et al., 2012), to the evolution of eddies in the ocean (e.g., Yang et al., 2024a) and seasonal and interannual dynamics (Pardo et al., 2019; Shadwick et al., 2023). Thus, a complete understanding of these processes requires high-frequency, and necessarily, automated observations, sustained over many years.

The SOTS observatory is comprised of two deep-water moorings: the Southern Ocean Flux Station (SOFS) focuses on heat, oxygen and $CO_2$ fluxes across the air-sea interface as well as the physical conditions and biological processes that control them (e.g. Schulz et al., 2012; Shadwick et al., 2015; Pardo et al., 2019; Yang et al., 2024a); the Subantarctic Zone (SAZ) sediment trap mooring focuses on quantifying the transfer of carbon to the ocean interior by sinking particles (i.e., the biological carbon pump; Trull et al., 2001b), providing samples for ecological (e.g., Trull et al., 2019; Rigual-Hernández et al., 2020a), paleo-proxy (e.g., King and Howard, 2003; Moy et al., 2009), and carbon flux estimates (e.g., Trull et al., 2001b; Wynn-Edwards et al., 2020c). Here we present observations collected at the SOTS site between 1997 and 2022 (Fig. 1 and Table A1). We construct seasonal climatologies of physical, chemical, and biological variables, including the inorganic carbon ($CO_2$) system, and quantify carbon export and its composition. We report here, for the first time, the full seasonal cycle of the phytoplankton (including coccolithophorid taxa), resolved from near weekly samples, and presented alongside the seasonal cycles of key hydrographic and biogeochemical parameters. Using records for a suite of physical and biogeochemical parameters that now range in length from $\sim 10$ years to more than $>20$ years, we evaluate changes in time. Finally, we provide a review of the research based on observations collected at the SOTS site over its lifetime, via both the mooring platforms themselves, and the associated annual deployment voyages, and offer perspectives for the ongoing value of fixed, open ocean, time-series initiatives.

## 1.1 Oceanographic Setting

The SOTS site is located in the SAZ southwest of Tasmania with the nominal location at 47°S, 142°E, within the area bounded by 46-48°S and 140-144°E (Fig. 1). The site is in a low current region, south of the Subtropical Front (STF), and in deep waters (>4500 m) west of the Tasman Rise. Observations from the SOTS site are representative of a broader region of the SAZ (e.g. Trull et al., 2001a; Shadwick et al., 2015; Trull et al., 2019; Yang et al., 2024a). There is a seasonal evolution of biomass accumulation that progresses from north to south (Fig. 2), as well as the presence of elevated chlorophyll on the Tasman shelf, northeast of the SOTS site. The accumulation of chlorophyll is relatively low, and generally distributed uniformly in the mixed-layer, unlike the conditions further south (i.e., south of the Subantarctic Front, SAF) where a subsurface chlorophyll maximum is present (e.g., Parslow et al., 2001; Boyd et al., 2024).

The upper water column at the site is characterised by warm, salty, and macro-nutrient poor, Subtropical water, from ex-tension of the Zeehan Current (Cresswell, 2000) and waters of the East Australian Current that transit through channels in the Tasman Rise (Fig. 2; Herraiz-Borreguero and Rintoul, 2011). This is underlain by cooler, fresher, and relatively oxygen-rich Subantarctic mode water (SAMW; Herraiz-Borreguero and Rintoul, 2010), which is formed by deep autumn and winter convection in the region (e.g. Tamsitt et al., 2020), and makes an important contribution to the uptake and storage of anthro-pogenic heat and $CO_2$ (Metzl et al., 1999; Sabine et al., 2004; Takahashi et al., 2009). Below the SAMW (Fig. 3), is cooler,

and more saline Antarctic Intermediate Water (AAIW), and below a depth of roughly 1500 m the water column is dominated by circumpolar deep water (CDW; Rintoul and Bullister, 1999; Traill et al., 2024).

## 2 Methods

### 2.1 The Southern Ocean Flux Station and Subantarctic Zone Moorings

All moorings are numbered sequentially (see Table A1 for deployment details), and are deployed and recovered annually on voyages to the SOTS region described below. Prior to 2010 a third mooring, the PULSE mooring, was deployed and used to measure upper ocean biogeochemistry. In 2016, the PULSE and SOFS moorings were combined and the SOFS prefix retained. Surface data from the SOFS mooring are relayed by satellite, while the subsurface data on both SOFS and SAZ are stored and downloaded when the moorings are recovered approximately 12 months later. All data are available via the Australian Ocean Data Network (AODN) Portal.

The SOFS mooring (Fig. A1) is a heavy-gauge 'S-tether' mooring with a meteorological tower that supports twin ASIMET systems measuring long and short-wave radiation, sea-level pressure, atmospheric temperature, humidity, rain and wind, all components necessary to quantify fluxes of heat, mass and momentum (Schulz et al., 2012), as well as photosynthetically active radiation (PAR). Surface wave spectra are measured using a motion reference unit. The partial pressure of $CO_2$ (p$CO_2$) in air and water is measured with an infra-red gas spectrometer (Sutton et al., 2014), and mixed-layer zooplankton and mesopelagic fish abundances estimated with a downward-looking acoustic backscatter instrument (Simrad WBat Mini). Finally, the SOFS surface float houses a modified McLane RAS-500 water sampler which collects 48 500 mL samples for analyses of nitrate, phosphate, silicate (Davies et al., 2020), and total alkalinity (Shadwick et al., 2020), as well as phytoplankton community composition by microscopy (Eriksen et al., 2020). We note that prior to SOFS-7, the RAS sampler was deployed at a depth of 30 m (Eriksen et al., 2018). A package of biogeochemical sensors (dissolved oxygen, chlorophyll fluorescence, optical backscatter) is located at 30 m, and a novel trace-metal clean water sampler located at 50 m (van der Merwe et al., 2011). Subsurface temperature, salinity and dissolved oxygen, are measured to quantify mixed-layer dynamics and capture the deep convection that occurs in winter (Weeding and Trull, 2014; Shadwick et al., 2015, 2023) The subsurface instruments are mounted at static depths (30m, 50m, 100m, 125m, 150m, 200m, 250m, 300m, and 500m in recent years; see Fig. A1). The SOFS mooring design has proven robustness at the SOTS site, with durations greater than 16 months with >90% data return. Fatigue calculations now suggest greater than four years durability for all elements of the mooring.

The SAZ sediment trap mooring (Fig. A2) is a sub-surface, high-tension, mooring equipped with three McLane PARFLUX-Mark78H conical traps (at nominally 1000, 2000, and 3800 m depth) that collect sinking particle flux samples with fortnightly resolution, preserved with mercuric chloride (Trull et al., 2001b; Wynn-Edwards et al., 2020c). Samples are analysed for total mass flux, particulate organic carbon (POC), particulate inorganic carbon (PIC) and biogenic silica (BSi) with details given in Wynn-Edwards et al. (2020b). The SAZ mooring also provides current meter measurements and a deep (∼4500 m) ocean conductivity, temperature and depth (CTD) sensor to measure ocean heat content below the depth of typical Argo profiling

float measurements (Jansen et al., 2022c). The SAZ mooring design is robust and SAZ deployments have performed faultlessly with 100% sample return between 2009 and 2021.

## 2.2  Annual Voyages and Ancillary Data

The annual deployment and recovery voyages to the SOTS site, on either the *RV Southern Surveyor,* or from 2015 onwards, the *RV Investigator* (see Fig. 4 and Table A1) collect additional sensor and sample measurements using various shipboard systems (e.g., a CTD rosette, nets, underway sensors, profiling floats, gliders, towed vehicles). Voyages generally occur in the austral autumn season (between March and May) partly due to ship logistics and partly to limit biofouling of sensors in advance of the initiation of the productive season. The full suite of activities undertaken on any particular voyage are detailed in individual

voyage reports which are produced annually and are available in an online repository (e.g., Wynn-Edwards et al., 2020a).

Ancillary data were used here to expand the observations beyond those measured by the SOTS moorings directly. The Ocean-Colour Climate Change Initiative (OC-CCI; Sathyendranath et al., 2019) product was used to make seasonal maps of surface chlorophyll concentration in the SOTS region (Fig. 2). A time-series of net primary production (NPP) from two satellite-based algorithms, the Vertically Generalized Production Model (VGPM; Behrenfeld and Falkowski, 1997) and the Carbon-based

Production Model (CbPM; Westberry et al., 2008), was sourced from the Oregon State Ocean Productivity website. The STF was defined by a potential temperature of $11°C$ at 150 m (Orsi et al., 1995), and computed using gridded Argo data with a spatial resolution of 1 degree (Roemmich and Gilson, 2009). Here, the time-series of STF location was averaged in two longitudinal ranges, 140-144°E, and, 130-150°E, to reflect the positions of the SOTS site, and broader region, respectively.

## 2.3  Data Quality Control Procedures, Construction of Seasonal Cycles, and Analysis of Trends

All sensor data collected at the SOTS site undergoes rigorous quality control (QC) within approximately 12 months of mooring recovery. The SOTS QC procedures apply automated tests following QARTOD recommendations for in-situ temperature, salinity, and oxygen data quality control, with the test parameters tailored to reflect regional oceanography. Additional detail for the temperature QC is given in Jansen et al. (2022c), for salinity in Jansen et al. (2022b), and for oxygen in Jansen et al. (2023). The QC procedure for the SOTS chlorophyll fluorescence (Chl) and optical backscatter ($B_{bp}$) sensor data are given in

Schallenberg et al. (2017). The QC of surface water and atmospheric $CO_2$ partial pressure ($pCO_2$) are undertaken by NOAA PMEL as described in Sutton et al. (2014).

Sample data from discrete observations collected at the SOTS site are also subjected to rigorous QC, generally within 12 months of mooring recovery. Analysis of nutrients (nitrate, phosphate and silicate) is undertaken at CSIRO following established procedures (Rees et al., 2019); detailed QC procedures for the nutrient observations are given in Davies et al.

(2020). Analysis of dissolved inorganic carbon ($TCO_2$) and total alkalinity (Alk) from discrete samples collected on annual voyages, and for samples collected on the SOFS mooring, is undertaken at CSIRO following Dickson et al. (2007) with QC procedures described in Shadwick et al. (2020). A relationship between Alk and salinity at the SOTS site has been developed from samples collected over many years (e.g., Shadwick et al., 2015, 2023), which is used to generate a time-series of Alk from sensor salinity. The computed Alk is then paired with the $pCO_2$ observations to compute time-series of surface ocean

TCO$_2$, pH, and aragonite saturation state ($\Omega$), using the CO2sys program (Lewis and Wallace, 1998; van Heuven et al., 2011; Sharp et al., 2023), the carbonate dissociation constants of Lueker et al. (2000), the fluoride dissociate constant of Perez and Fraga (1987), the sulphate dissociate constant of Dickson (1990), and the boron to salinity ratio of Lee et al. (2000). Analysis of samples collected from the sediment traps are undertaken at the Institute for Marine and Antarctic Studies at the University of Tasmania; both analytical methods and QC procedures are described in Wynn-Edwards et al. (2020b).

A gridded data set of temperature in the upper 500 m was constructed from averaging values acquired within 30 minutes of each hour at each depth; additional details on the construction of the gridded data are given in Jansen et al. (2022a). Mixed-layer depth (MLD) is computed from profiles of temperature following Weeding and Trull (2014) and Shadwick et al. (2015). Seasonal cycles of sensor data (temperature, salinity, oxygen, Chl, B$_{bp}$, pCO$_2$ and associated carbonate system parameters) and sample data (nitrate, silicate, and POC, PIC, and BSi at 2000 m) were constructed by combining data collected in all years to establish mean monthly values for each parameter. Note that the SAZ-11, SAZ-19, and SOFS-6 data have been included in the gridded temperature product described above, but excluded in the construction of subsequent seasonal climatologies because the moorings were deployed north of the nominal SOTS site (see Table A1).

The QC procedures for the phytoplankton samples are described in Eriksen et al. (2020). Sub-samples are quantitatively analysed for coccolithophore (including calcification morphotypes Rigual-Hernández et al. (2020a, b); Shadwick et al. (2021)) community composition by Scanning Electron Microscopy (SEM; see Fig. A3) and analysis of the broader micro-planktonic community by Light Microscopy (LM). Because sample return has been intermittent (Eriksen et al., 2020), seasonal cycles presented here are constructed by combining observations from both the PULSE mooring, and the later SOFS configuration (Table A1).

The detection of trends in several of the SOTS records was undertaken following Sutton et al. (2022) using the Trends of Ocean Acidification Time Series (TOATS) open-source code. This method requires records with a similar number of observations in each month of the year (i.e., uniformly distributed over the annual cycle), which restricted the analysis to the following parameters: nitrate, silicate, pCO$_2$, MLD, temperature, salinity, NPP, POC, PIC, BSi (see Table 2).

## 3 Results

### 3.1 Upper Ocean Seasonality

The SOTS site is characterised by deep mixing in the autumn and winter seasons, in some years to depths of roughly 500 m (Fig. 5b), driven by a combination of local heat fluxes (Schulz et al., 2012), and northern Ekman transport of colder waters (Rintoul and England, 2002). This deep mixing is associated with the seasonal formation of SAMW (e.g. Tamsitt et al., 2020). Stratification in spring yields mixed layers of 50 to 100 m depth, which persist until the following autumn largely due to sustained wind speeds of $\sim$10 m s$^{-1}$ (e.g., Trull et al., 2019).

While the upper ocean heat content and associated mixed-layer depth is driven by the annual cycle of incoming solar radiation (Schulz et al., 2012), large variations on hourly to daily timescales are dominated by the horizontal advection of water past the SOTS site (Pardo et al., 2019; Yang et al., 2024a). This highlights the Eulerian nature of the observations and the

importance of understanding spatial context, including movements of the dynamically active SAF to the south, which bounds the Antarctic Circumpolar Current, and the STF to the north, a weaker but more variable feature, which contributes to flows of warm waters westward south of Australia. The position of the STF varies seasonally and inter-annually, at times migrating south of the nominal SOTS site (Fig. 5a). Depending on the longitudinal range considered, the STF may be observed as far south as 48°S. During the winter and spring, the STF is located further north, near the northern boundary of the SOTS region (Fig. 5a). Because the SOTS site is in the northern part of the SAZ (i.e., much closer to the STF than the SAF), variability in the surface ocean is more strongly influenced by variability in the location of STF. However, earlier work has shown that advection of waters with more polar characteristics can influence properties at the SOTS site on sub-seasonal timescales (Shadwick et al., 2015).

The temperate conditions at the SOTS site result in surface water temperatures ranging from summer maxima of 15°C to winter minima of ∼8°C (Fig. 5b). The mean magnitude of the seasonal temperature cycle at the surface is 4°C, with summer maximum of roughly 12°C; the climatological seasonality smooths some of the variability observed in the higher frequency records. At depths below 200 m, the magnitude of the seasonal variation is much smaller (Figs. 6 and 7a).

The seasonal variation in surface water salinity is modest (Fig. 7b), though large departures from the mean conditions can occur, with maxima approaching 35.5, and minima approaching 34 (Fig. 8a), reflecting the episodic passage of Subtropical water from the north and Subantarctic waters from the south (Shadwick et al., 2015; Pardo et al., 2019), or the impact of mesoscale features (Yang et al., 2024a). Below a depth of roughly 200 m, seasonal variability in salinity is small, with profiles converging at roughly 34.6 (Fig. 7b), reflecting the mixture of Subtropical Surface Water (STCW) and underlying SAMW (Fig. 3). Seasonal variability in dissolved oxygen reflects the changes in hydrography (temperature and salinity), as well as biological processes, with generally elevated values at the surface in the spring and summer seasons, relative to winter conditions, and lower subsurface concentrations in all seasons (Fig. 7c).

## 3.2 The $CO_2$ System at the SOTS site

The seasonality of the carbonate system in the surface waters at the SOTS region is controlled by a combination of physical and biological drivers. The seasonality in temperature is moderate as described above, and the seasonal cycle of $pCO_2$ (Fig. 8b) is dominated by the impact of biological productivity, and the introduction of inorganic carbon ($TCO_2$) rich waters from below (Metzl et al., 1999; Shadwick et al., 2015, 2023). This is reinforced by the seasonality of $O_2$ saturation, which is anti-correlated with $pCO_2$ throughout the year, with supersaturated conditions generally occurring between December and March (Fig. 8b), and undersaturated conditions in the autumn and winter seasons. The seasonality in alkalinity (Alk) is computed from salinity and thus exhibits similarly modest seasonality (Fig. 8a and c). The seasonality in $TCO_2$ (Fig. 8d) is consistent with that of $pCO_2$ with elevated values in the autumn and winter season influenced by a combination of biology (remineralisation of organic matter produced in the previous productive season) and the introduction of $TCO_2$-rich water from below during periods of deep mixing (Fig. 7e). The seasonality of pH and carbonate saturation state ($\Omega$) are anti-correlated with $pCO_2$ and $TCO_2$ (Fig. 8d-f).

While the variability of $TCO_2$ in the surface ocean is controlled by a combination of biological (photosynthesis and respiration, and the formation of calcium carbonate, $CaCO_3$), and physical (evaporation and precipitation, sea ice formation and melt, advection, air-sea gas exchange, and $CaCO_3$ dissolution) processes, the variability at depth is controlled primarily by the remineralisation of organic matter, and circulation, or mixing. The subsurface concentrations of $TCO_2$ at the SOTS site range from roughly 2100 $\mu$mol kg$^{-1}$ at 500 m to greater than 2250 $\mu$mol kg$^{-1}$ at depths below 1500 m (Fig. 9a). The elevated concentrations in the interior ocean reflect both the remineralisation of organic carbon and the dissolution of $CaCO_3$. Like $TCO_2$, the variability of Alk in the surface ocean is controlled by the addition (precipitation and sea ice melt) and removal (evaporation and sea ice formation) of freshwater, which also change salinity. The formation and dissolution of $CaCO_3$, as well as photosynthesis and respiration, also impact surface Alk, though the former has a much larger impact. In the subsurface, Alk is influenced by mixing and $CaCO_3$ dissolution and concentrations of Alk range from roughly 2280 $\mu$mol kg$^{-1}$ at 500 m to greater than 2350 $\mu$mol kg$^{-1}$ below 1500 m (Fig. 9b). The relationship between Alk and salinity at the SOTS site reveals two quasi-linear relationships: a high slope relationship in Alk-rich, deep ($>$800 m), waters over a small range in salinity, and a lower slope relationship in shallow (surface to roughly 800 m depth) waters over a much wider range of salinity, and with relatively lower Alk (Fig. 9c), which reflects the water column structure with Subtropical surface waters overlying mode and intermediate waters (Fig. 3).

### 3.3 Primary Production and Phytoplankton Community Composition

While biological productivity has been shown to dominate the seasonality in the carbonate system and influence the air-sea flux of $CO_2$ in the region (Shadwick et al., 2015, 2023; Yang et al., 2024a), the SOTS site exhibits the high-nutrient, low-chlorophyll (HNLC) characteristics of the broader SAZ region (Fig. 2). Observations of both chlorophyll and optical backscatter from sensors on the SOFS mooring are available as proxies for primary production (Schallenberg et al., 2019; Yang et al., 2024a), with seasonal cycles indicating that the onset of the productive season generally occurs in early austral spring, evident in the concurrent increase in both parameters (Fig. 10a) though the timing of this initiation has been shown to vary from year to year (e.g. Weeding and Trull, 2014; Trull et al., 2019). Productivity peaks between January and March (summer), based on both chlorophyll observations (Fig. 10a) and satellite-derived estimates of net primary production (NPP, Fig. 10c), with biomass accumulation continuing through June (winter) in most years. The mean NPP at the SOTS ranges from 56 to 130 mg C m$^{-2}$ y$^{-1}$ depending on the choice of productivity model (see Tables 1 and A2). The major limitation on primary productivity appears to be iron (e.g. Sedwick et al., 1999), with nitrate concentrations remaining high year-round, while silicate becomes seasonally depleted in summer (Fig. 10b). Nutrients are anti-correlated with biomass (Fig. 10a,b); silicate utilisation begins in early spring, with limitation reached by January, and ongoing productivity fueled by regenerated nutrients (Lourey and Trull, 2001; Trull et al., 2019).

The micro-plankton community at the SOTS site has a subdued but distinct seasonal cycle in both abundance (Fig. 11a), and the diversity of silicifying (diatoms, silicoflagellates, some radiolaria), calcifying (forams and coccolithophores) and other (flagellates and dinoflagellates) taxa (Figs. 11b and A3). The phytoplankton community is numerically dominated by small cells, specifically, the flagellate form of *Phaeocysistis antarctica* and lesser contributions from *Phaeocysistis scrobiculata*.

Flagellates, silicoflagellates, foraminifera and radiozoa are relatively minor contributors to biovolume (Fig. 11b), which is dominated by diatoms and dinoflagellates. Late spring, and summer samples are often dominated by large-biovolume taxa like the diatom, *Corethron pennatum*, before reverting to a more diverse community of smaller diatom taxa. However, diatoms are a relatively small component of the total community, with a modest seasonal increase in abundance in mid-summer after the onset of silicate limitation.

All major phytoplankton groups are present throughout the year (see Figs. 11a and 12) with subtle community composition changes likely tied to the distinct stages in the development of the mixed-layer: deep mixing and light limitation in early spring; shoaling in late-spring or early summer before the onset of nutrient limitation; and an extended period of stability in summer and early autumn. It is interesting to note that the silicoflagellates do not mimic the diatom seasonality (Fig. 12b). We note that the total abundances of the silicoflagellates are low, typically 2 orders of magnitude lower than that of the total diatoms; it

may be that the diatoms outcompete the silicoflagellates for available silica in summer, contributing to their low abundances. Additionally, silicoflagellates are only confidently identified during the part of their life-cycle when siliceous exoskeletons are produced; naked stages are poorly resolved and only recently linked to the skeleton-bearing stages (McCartney et al., 2024).

There are no strong signals in community composition associated with the migration of the STF past the SOTS site; however these conditions are typically short-lived (Fig. 5a) and the presence of indicator taxa may be too low to be clearly resolved.

There is some evidence of seasonal succession between major functional groups, with most groups low in winter, coccolithophore abundance highest in spring, followed by increased abundance of dinoflagellates, flagellates and diatoms in summer (Fig. 12).

The seasonality of species richness at the SOTS site (Fig. 11c) closely follows total abundance (Fig. 11a), chlorophyll (Fig. 10a) and NPP (Fig. 10c), with minima in winter and and highest values observed during the productive season, or (austral)

summer months. The total number of taxa recorded at the SOTS site since the inception of taxonomic sampling is over 300, and is a result of the combination of both SEM and LM examination.

### 3.4 Particulate Carbon Export

The movement of carbon to the ocean interior by sinking particles, the gravitational biological carbon pump, is active year round at the SOTS site ($1.4 \pm 0.3$ g m$^{-2}$ yr$^{-1}$ at 2000 m) and occurs at levels similar to the global median ($\sim 1$ g m$^{-2}$ yr

$^{-1}$) (Wynn-Edwards et al., 2020c; Lampitt and Antia, 1997; Mouw et al., 2016). The flux of particulate organic carbon at 2000 m indicates an increased delivery of material in September, with peak fluxes ($\sim$6 mg C m$^{-2}$ d$^{-1}$) occurring in mid-summer (December and January), and a steady rain of particulate carbon through the winter season (e.g., mean monthly fluxes in winter are on the order of 2 mg C m$^{-2}$ d$^{-1}$; Fig. 10c). This seasonality is well correlated with satellite-based estimates of NPP and confirms that there is little, if any, lag between the onset of production at the surface and the delivery of organic carbon to depth

(Fig. 10c; Wynn-Edwards et al., 2020c).

The composition of particulate material delivered to depth reveals a similar seasonality as the POC flux, with a steady increase in both particulate inorganic carbon (PIC, or CaCO$_3$) and biogenic silica (BSi, or Opal) beginning in September, with peak fluxes ($\sim$8 mg m$^{-2}$ d$^{-1}$ for both PIC and BSi) in summer (Fig. 10d, Table 1). The PIC seasonality is characterised by

two distinct peaks (in spring and summer), while the BSi seasonality is broader, possibly due to the merging, or overlapping, of peaks in spring and summer. The amount of material arriving at depth is determined by two factors: (1) how much of the available particle pool is exported out of the surface layer (i.e., the export ratio) and; (2) what fraction of that export arrives at depth (i.e., the transfer efficiency; Henson et al., 2012). Annual transfer efficiency at the SOTS site, calculated from NPP (Fig. 10c) using export efficiency estimates following Britten et al. (2006) and annual POC fluxes at 2000 m (1.37 mg mg C m$^{-2}$ d$^{-1}$), ranges from 4 to 5%, depending on the choice of NPP product (VGPM or CbPM), and is consistent whether computed using the POC flux at 2000 m (Fig. 10c) or 1000 m (Table A2).

### 3.5 Emergence of Trends

Many of the records acquired at the SOTS site are now of sufficient length that they can be used to evaluate the emergence of trends; the results of the trend analysis are summarised in Table 2. The increase in surface water $pCO_2$, along with an apparent amplification of the seasonality, have been observed at the SOTS site and are described in detail in Shadwick et al. (2023). This increase in $pCO_2$ corresponds to both the increase in atmospheric $CO_2$, and a decrease in surface ocean pH (i.e., ocean acidification), and reinforces the need for ongoing observations to separate natural variability from anthropogenic trends (e.g. Ono et al., 2019). The satellite-based estimates of NPP indicate modest trends (on the order of -8 mg C m$^{-2}$ y$^{-1}$), indicating small decreases in NPP in the SOTS region over about a decade, that while statistically significant, are associated with fairly small $R^2$ (Table 2), indicating that the trend explains only a small fraction of the variability in the records. There is also a weak, but significant, trend in silicate ($\sim$0.1 $\mu$mol kg$^{-1}$ y$^{-1}$), potentially indicating a decline in nutrient utilisation, which is consistent with a decline in NPP, though no trend was detected in the time-series of nitrate. None of the other surface or subsurface parameters, including the hydrography, the particulate flux and the composition of the exported material, indicated any trends over the history of the SOTS observations.

## 4  Discussion

The SOTS observatory is the only sustained multi-disciplinary, high-temporal resolution, observational effort in open Southern Ocean waters. The observations collected at the SOTS site thus serve as an important baseline against which to measure future changes. Additionally, observations collected on annual voyages, from ancillary autonomous platforms, and remote sensing in the SOTS region underpin research conducted across a suite of Australian programs and international initiatives, examples of which are provided in the following sections.

### 4.1  Upper Ocean biogeochemistry at the SOTS site

A central theme of the SOTS science program is an assessment of the key processes responsible for variability in upper ocean biogeochemistry. Early efforts highlighted the importance of biology on the seasonality of the inorganic carbon system (Shadwick et al., 2015) and quantified internanual $CO_2$-system variability (Pardo et al., 2019). Recent work expanded the surface water $pCO_2$ time-series via a multiple linear regression and remote sensing observations to evaluate the impact of the

Southern Annular Mode and mesoscale circulation on surface water $CO_2$ seasonality (Yang et al., 2024a). Using observations collected over a decade, an amplification of the seasonal cycle in surface water $pCO_2$ was quantified and associated with a decline in pH as a consequence of the uptake of anthropogenic $CO_2$ (Shadwick et al., 2023).

Net community production (NCP), which sets the upper limit for carbon export to the ocean interior, has been computed from seasonal budgets of oxygen (Weeding and Trull, 2014; Trull et al., 2019) and inorganic carbon (Shadwick et al., 2015) from moored sensors, and from seasonal nitrate deficits using discrete samples (Lourey and Trull, 2001). Annual NCP at the SOTS site ranges from 1.5 to 3.4 mol C m$^{-2}$ yr$^{-1}$; the NCP that occurs during winter and early spring accounts for roughly 30% of the annual total. This appears to result from reduction of grazing in winter (Trull et al., 2019), suggesting future changes in the SAZ may be controlled as much by winter as by summer conditions.

The seasonal succession of the phytoplankton community has also been elucidated on the basis of chlorophyll and backscatter data (Schallenberg et al., 2019). A combination of shipboard incubation experiments and underway fast repetition rate fluorometer observations were used to evaluate the potential of non-photochemical quenching to indicate conditions of iron limitation (Schallenberg et al., 2020), a key constraint on biological productivity at the SOTS site and in the broader SAZ region (Bowie et al., 2009). More recently, shipboard incubation experiments at the SOTS site and elsewhere in the SAZ have shown that phytoplankton may be seasonally co-limited by iron and manganese (Latour et al., 2023), while deep mixing in autumn and winter also leads to light limitation (e.g. Trull et al., 2019).

The phytoplankton community composition and seasonal succession has been described for both the silicifying phytoplankton (i.e., diatoms; Eriksen et al., 2018), as well as the calcifying community (i.e., coccolithophores; Rigual-Hernández et al., 2020a) at the SOTS site. While the non-calcifying community is dominated by very small cells, there is a seasonal depletion of silicate in (austral) summer between January and April, and export of organic carbon reaches levels ($1.4 \pm 0.3$ g m$^{-2}$ yr$^{-1}$ at 2000 m) similar to the global median ($\sim 1$ g m$^{-2}$ yr$^{-1}$) (Wynn-Edwards et al., 2020c; Lampitt and Antia, 1997; Mouw et al., 2016). This suggests that natural increases in iron fertilization by the southward extension of the East Australian Current (e.g., Ridgway, 2007), or by atmospheric deposition (e.g., Traill et al., 2022), may drive increased carbon sequestration by the biological carbon pump, without limitation by silicate supply as previously expected (Trull et al., 2001a).

While the calcifying community has proven more challenging to observe from water samples collected autonomously at the SOTS site (Eriksen et al., 2020), the relative abundances of five morphotypes of the coccolithophore *Emiliania huxleyi* over a full annual cycle have been quantified (Rigual-Hernández et al., 2020a). An assessment of this seasonal succession with respect to the evolution of the biogeochemical and hydrographic properties in the upper ocean from mooring observations suggests that the heavier morphotypes achieve maximum abundance in winter, when nutrients and $TCO_2$ are elevated, but calcium carbonate saturation states are low (see also Fig. 8), while lighter, more weakly calcified morphotypes are dominant in the summer season (Rigual-Hernández et al., 2020a). Extensive genetic variability in the sampled population of *E. huxleyi* is likely to contribute to the response of the different morphotypes to seasonal changes in environmental conditions (Rigual-Hernández et al., 2020a). This is somewhat at odds with the view that ocean acidification, and the subsequent decrease in upper ocean carbonate ion concentration and carbonate saturation states, will lead to a shift away from heavily-calcified coccolithophores (Riebesell et al., 2000; Beaufort et al., 2011).

## 4.2 Carbon Export to the Deep Ocean

Early efforts at the SOTS site were focused on quantifying the gravitational biological carbon pump (Trull et al., 2001b; Boyd and Trull, 2007), while more recent work has also characterized the composition of particles exported to depth and quantified interannual and longer term variability (Wynn-Edwards et al., 2020c), as well as the role of additional particle injection pumps (Boyd et al., 2019) for the delivery of organic material to the ocean interior (Yang et al., 2024b; Thompson et al., 2024). Yang et al. (2024b) provide an independent estimate of particle export, based on optical sensors on profiling floats (BGC-Argo) and showed that 79% of export happens via gravitational sinking, which indicates that biological processes, either directly via production of faecal pellets and/or aggregates or vertical migration, or indirectly via remineralisation, are likely the main modulators of particle flux at depth in the SOTS region and in the broader SAZ.

In addition to quantification of carbon export, several studies have used the preserved sediment trap samples for investigations focused on diatoms (e.g., Closset et al., 2015; Rigual-Hernández et al., 2016; Wilks et al., 2017; Wilks and Armand, 2017) and calcifying phytoplankton (e.g., King and Howard, 2003; Moy et al., 2009; Rigual-Hernández et al., 2020b), as well as seasonal variations in acantharia and their contribution to particulate export (Sun et al., 2024), and to quantify lithogenic particle fluxes to depth via atmospheric dust deposition (Traill et al., 2022). Analyses of coccolithophore samples collected at the SOTS site have shown that *E. huxleyi* is the most abundant, but does not dominate in terms of the amount of carbonate that it precipitates and subsequently exports to depth (Rigual-Hernández et al., 2018, 2020b). This challenges the view that *E. huxleyi* is the dominant species with respect to carbon export in the SAZ (Rigual-Hernández et al., 2020b), with larger, more heavily calcified taxa like *Calcidiscus leptoporus*, *Coccolithus pelagicus* and *Helicosphaera carteri* considered the 'heavy lifters'. Because the removal of carbonate from the upper ocean by calcifying plankton plays an important part in the ocean carbon cycle, these nuances are important to the assessment of ecosystem change and potential consequences of ocean acidification (e.g. Shadwick et al., 2021).

Sediment trap records from the SOTS site have also been used to evaluate changes in the planktonic community occurring over longer timescales. A unique comparison of foraminifera shells collected by the SAZ mooring and those preserved in the underlying sediments suggested that modern shells were substantially lighter, consistent with reduced calcification in the present day, relative to the Holocene (Moy et al., 2009). A similar analysis of *E. huxleyi* collected in surface water samples, sediment trap samples, and Holocene-era sediments revealed a much more subtle change in shell weight (Rigual-Hernández et al., 2020), while an additional study of a different coccolithophore, *Calcidiscus leptoporus,* indicated a considerable reduction in size in modern (sediment trap) samples (Rigual-Hernández et al., 2023). These studies highlight the complex and varying response of different calcifying phytoplankton to changes in their environment over these timescales.

## 4.3 Comparison with Other High-Latitude Time-Series

Time-series observations have advanced our understanding of a range of processes, from seasonal (e.g., Imai et al., 2002; Brix et al., 2013; Koelling et al., 2022) to interannual (e.g. Currie et al., 2011; Bates et al., 2014; Sutton et al., 2017) to longer term variability in the upper ocean (e.g. Ishii et al., 2011; Ono et al., 2019; Duke et al., 2023) and ocean interior (Honda

et al., 2002; Conte et al., 2025). There are relatively fewer observations in the southern high latitudes (e.g. Bates et al., 2014), and parallels are often drawn between the SAZ and the HNLC waters of the Subartic North Pacific (e.g. Wong and Matear, 1999; Shadwick et al., 2015). The annual cycles of surface water temperature at both Ocean Station Papa (OSP, in the eastern subartic North Pacific) and the K2 site (western subarctic North Pacific) are significantly larger than at the SOTS site, and while both sites undergo mixed-layer deepening in the winter season (to ∼100-150 m; McKinley et al., 2009; Wakita et al., 2010), neither exhibit the depths observed at the SOTS site. At OSP the seasonal amplitude in surface temperature is roughly 10°C (Sutton et al., 2017; Duke et al., 2023), and at K2 it is 15°C (Wakita et al., 2010). In both regions, this seasonal warming counteracts the biologically-driven decrease in surface water $pCO_2$, and limits the oceanic uptake of $CO_2$ during the productive season (Fassbender et al., 2016; Sutton et al., 2017), a phenomenon not observed at the SOTS site (e.g. Shadwick et al., 2015). Time-series of upper ocean nutrients and biomass at the KNOT station (near the K2 site in the western subarctic North Pacific) indicate the magnitude of seasonal nitrate drawdown is larger than at the SOTS site, and occurs without the characteristic seasonal silicate limitation of the SAZ (Tsurushima et al., 2002). The seasonal change in upper ocean $TCO_2$ at the KNOT site, may be more than double that observed at the SOTS site (Tsurushima et al., 2002; Wakita et al., 2010), potentially reflecting the more biologically productive summer conditions of the western North Pacific.

Estimates of net community production (NCP) at OSP ($2 \pm 1$ mol C m$^{-2}$y$^{-1}$; Fassbender et al., 2016) are comparable with values estimated for the SOTS site (Weeding and Trull, 2014; Shadwick et al., 2015), suggesting that organic carbon available for export to the ocean interior might be similar. Estimates of particulate flux at OSP however indicate greater delivery of material to depth, potentially as much as double the POC flux during peak periods, along with comparable but higher $CaCO_3$ flux, and significantly more Opal (or BSi; Timothy et al., 2013). These higher values of export, considered in the context of similar estimates of NCP at OSP suggest that the transfer efficiency in the Subarctic North Pacific is higher. The higher flux of Opal at OSP may be due to availability of nutrients year-round (e.g. Harrison, 2002), in contrast to the seasonally limiting concentrations of silicate in the SAZ (see Fig. 10), though the OSP region is seasonally Fe-limited which exerts some control on the growth of large diatoms (Harrison, 2002). Recent work has shown that in the Southern Ocean shallow mixed-layer depths are associated with less re-entrainment of exported POC (and the products of remineralised POC), leading to greater export (McClish et al., 2025). If this mechanism were to operate in the Subarctic North Pacific, the shallower winter mixed-layer may also contribute to the greater export at OSP.

## 5   Conclusions

The SOTS program began as a sediment trap time-series in 1997, and has since then expanded via partnerships with the Australian Integrated Marine Observing System (IMOS), the Australian Bureau of Meteorology, and most recently the Australian Antarctic Program Partnership to address surface ocean processes including air-sea fluxes (Schulz et al., 2012), surface waves (Rapizo et al., 2015), carbonate chemistry (Shadwick et al., 2023), biological productivity (Trull et al., 2019), and phytoplankton community composition (Eriksen et al., 2018) and biodiversity. The SOTS program is part of the OceanSITES network of fixed time-series stations, and contributes data to the Global Carbon Project, and the Surface Ocean Carbon Atlas. The remit of

the SOTS program is extended beyond the site to both broad and mesoscale understanding of Southern Ocean productivity and circulation via sensors designed for calibration and validation of in-ocean platforms (Wynn-Edwards et al., 2023; Yang et al., 2024b), and satellite altimetry (Ardhuin et al., 2024; Bohé et al., 2024), respectively.

Sustained time-series observations that resolve processes at timescales relevant to quantifying natural variability are essential to the detection and attribution of anthropogenic change (e.g., Bates et al., 2014; Sutton et al., 2019; Shadwick et al., 2023). By continuing the observational records acquired at the SOTS site, the progress of ocean acidification, deoxygenation, and availability of iron and other nutrients as inputs to broader assessments of expected changes in marine ecosystems can be made. Improved understanding of mechanisms controlling heat and carbon uptake by the ocean will be used to improve their representation in models and forecasts, which are the tools needed to provide advice about climate variability and its likely future impacts on biogeochemical cycles and marine ecosystems.

*Data availability.*  The observations collected at the SOTS site can be obtained from the Australian Ocean Data Network (AODN; IMOS, 2022) portal: https://portal.aodn.org.au/. The SOTS voyage annual reports are available at from the IMOS SOTS webpage (https://imos.org.au/facility/deep-water-moorings/southern-ocean-time-series-observatory).

*Author contributions.*  EHS and CWE conceived the study; EHS, CWE, PJ, DD, RE, and ES contributed to data acquisition, data QC, and analysis. EHS, CWE and RE wrote the manuscript. All authors contributed to editing of text and production of figures.

*Competing interests.*  The authors declare no competing interests.

*Acknowledgements.*  The SOTS observatory is supported by Australia's Integrated Marine Observing System (IMOS). IMOS is operated by a consortium of institutions as an unincorporated joint venture, with the University of Tasmania as Lead Agent. The SOTS observatory is part of the OceanSITES global network of time-series stations (www.OceanSITES.org). This research was supported by a grant of sea time on RV *Investigator* from the CSIRO Marine National Facility (https://ror.org/01mae9353), and by grant funding from the Australian Government as part of the Antarctic Science Collaboration Initiative program. We thank staff at the University of Tasmania Central Science Lab (K. Goemann, S. Feig) for expert assistance with SEM. The SOTS Team is grateful for the heroic efforts of Tom Trull who started the program and ensured its continuity for many, many, years.

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

**Table 1.** Annual mean (g m$^{-2}$ y$^{-1}$), peak (Dec), and mean winter (JJA; g m$^{-2}$ yr$^{-1}$) particle fluxes at 2000m, and annual mean NPP (mg C m$^{-2}$ yr$^{-1}$), peak and mean winter NPP (mg C m$^{-2}$ d$^{-1}$).

|  | Total mass | POC | PIC | BSi | NPP(VGPM) | NPP(CbPM) |
|---|---|---|---|---|---|---|
| Annual | 21.8 | 1.4 | 1.9 | 1.2 | 130.4 | 56.5 |
| Peak (Dec) | 104.3 | 6.6 | 8.3 | 7.8 | 680.1 | 357.6 |
| Winter Mean | 38.3 | 2.2 | 3.2 | 1.5 | 161.8 | 15.93 |

**Table 2.** Results of the trend analysis for parameters measured at the SOTS site with sufficiently long records, with the resulting trends given in the units shown per year. The nitrate, silicate, and pCO$_2$ observations are from samples collected at the surface. The surface temperature and salinity are from observations at 30 m depth. There were no significant trends in any of the subsurface (below 50 m) temperature or salinity records. The POC, PIC, and BSi entries refer to observations collected by the 2000 m sediment trap; despite these being the longest records in the time-series, they do not exhibit any trends.

| parameter (units) | record length | trend $\pm$ uncert. | $R^2$ | slope coef. |
|---|---|---|---|---|
| Nitrate ($\mu$mol kg$^{-1}$) | 12.7 | — | — | — |
| Silicate ($\mu$mol kg$^{-1}$) | 11.7 | $0.08 \pm 0.02$ | 0.27 | $p < 0.05$ |
| pCO$_2$ ($\mu$atm) | 10.5 | $2.63 \pm 0.28$ | 0.56 | $p < 0.05$ |
| MLD (m) | 12.3 | — | — | — |
| surface temp. ($^\circ$C) | 11.7 | — | — | — |
| surface salinity (kg m$^{-3}$) | 11.7 | — | — | — |
| NPP VGPM (mg C m$^{-2}$ y$^{-1}$) | 10.2 | $-8.0 \pm 2.7$ | 0.08 | $p < 0.05$ |
| NPP CbPM (mg C m$^{-2}$ y$^{-1}$) | 8.3 | $-8.4 \pm 2.5$ | 0.14 | $p < 0.05$ |
| subsurface temp. ($^\circ$C) | 12.2 | — | — | — |
| subsurface salinity (kg m$^{-3}$ ) | 12.2 | — | — | — |
| Total Mass (mg m$^{-2}$ d$^{-1}$) | 24.6 | — | — | — |
| POC (mg m$^{-2}$ d$^{-1}$) | 24.6 | — | — | — |
| PIC (mg m$^{-2}$ d$^{-1}$) | 24.6 | — | — | — |
| BSi (mg m$^{-2}$ d$^{-1}$) | 24.6 | — | — | — |

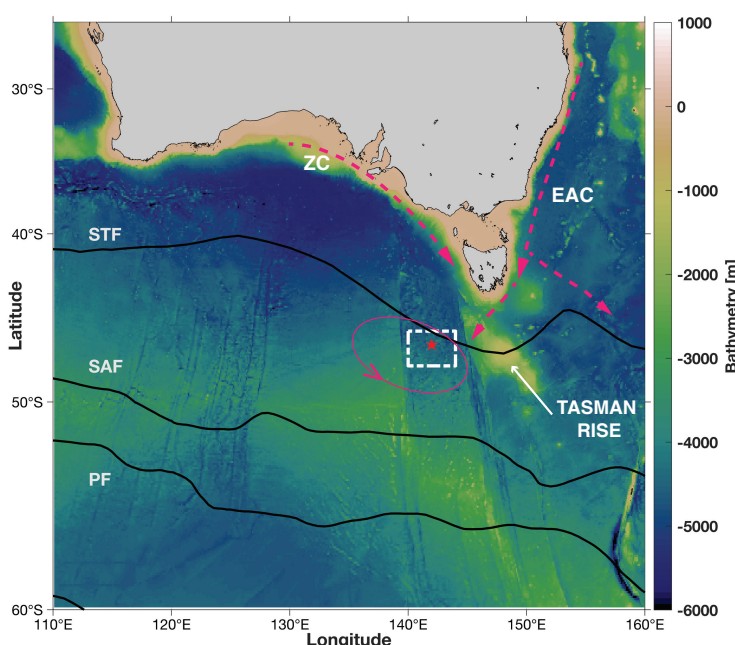

**Figure 1.** The location of the SOTS site (red star) and broader region (white box) in the Subantarctic zone (SAZ) southwest of Tasmania. Also shown are the climatological locations of the Subtropical (STF), Subantarctic (SAF), and Polar (PF) fronts following Orsi et al. (1995), and the schematic positions of the Zeehan Current (ZC), extensions of the East Australian Current (EAC), and a recirculation west of the Tasman Rise following (Herraiz-Borreguero and Rintoul, 2010). The bathymetry is from the GEBCO 2024 gridded product.

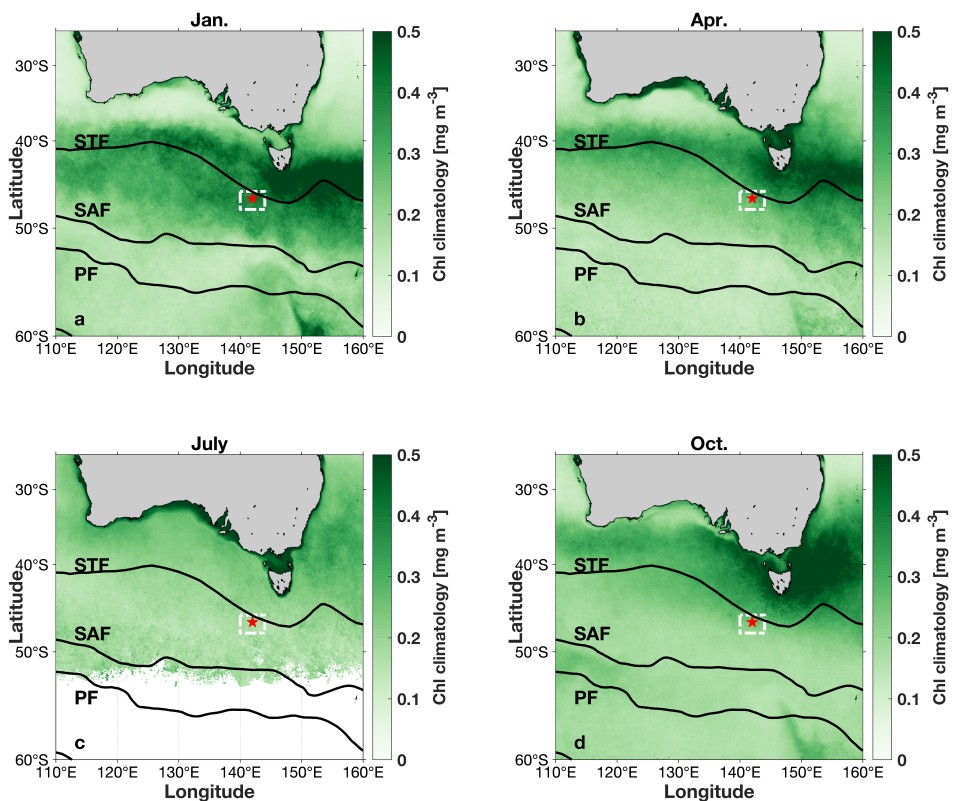

**Figure 2.** Mean seasonal chlorophyll concentrations from satellite observations in the SAZ region south of Australia. The SOTS site (red star) and broader region (white box) are shown, along with the climatological locations of the STF, SAF, and PF as in Fig. 1. a) January (summer), b) April (autumn), c) July (winter), and d) October (spring).

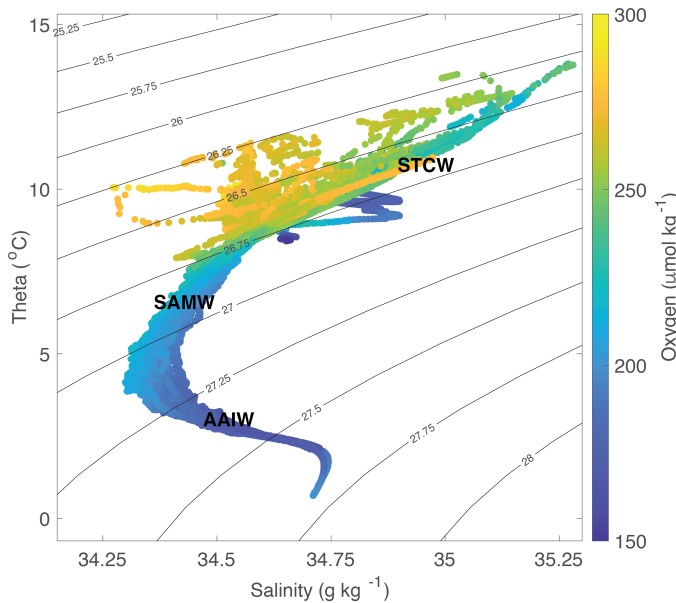

**Figure 3.** Temperature versus salinity for all shipboard CTD sensor data collected between 2010 and 2022 at the SOTS site, with the concentration of dissolved oxygen indicated by the colour, and the major water masses: Subtropical Surface Water (STCW), Subantarctic Mode Water (SAMW), and Antarctic Intermediate Water (AAIW) indicated schematically.

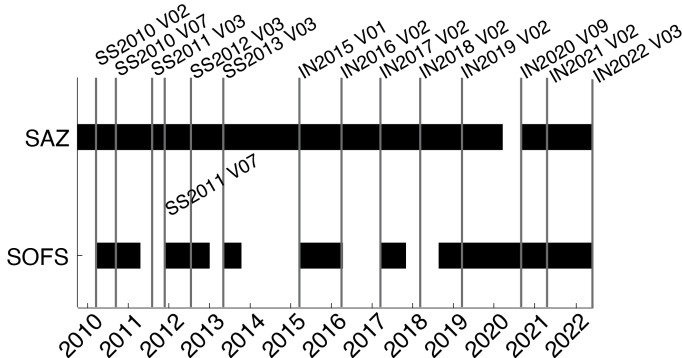

**Figure 4.** History of mooring deployments at the SOTS site between 2010 and 2022 and the associated voyage names, with the prefix 'SS' referring to the *RV Southern Surveyor* and the prefix 'IN' referring to the *RV Investigator*. The SAZ mooring record goes back much further, beginning in 1997 (see Table A1 for additional details).

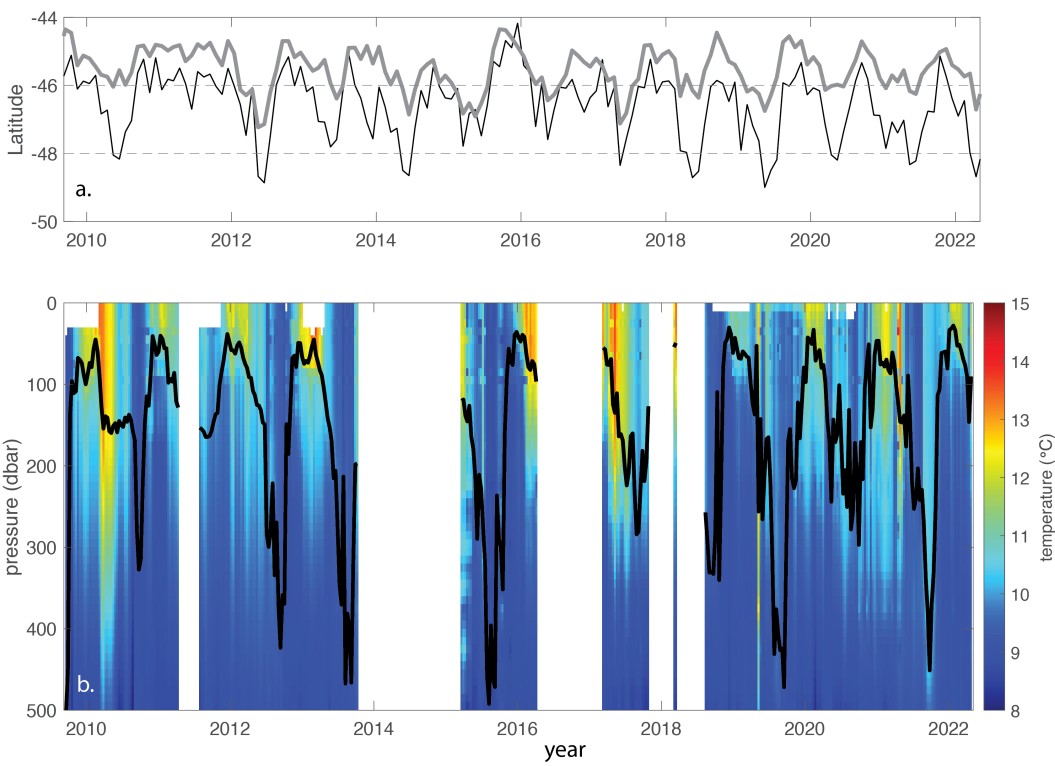

**Figure 5.** The position of the STF and upper ocean temperature between 2010 and 2022. a) the location of the STF in two longitudinal ranges: 140-144°E (thin black line), and 130-150°E (thick line) is compared to the location of the SOTS site (between 46°S and 48°S, dashed horizontal lines). b) Temperature in the upper 500 m of the water column from sensors on the SOFS moorings, and the associated mixed-layer depth (black line, defined as a temperature difference of 0.3°C from a depth of 10 m) from observations collected between 2010 and 2022.

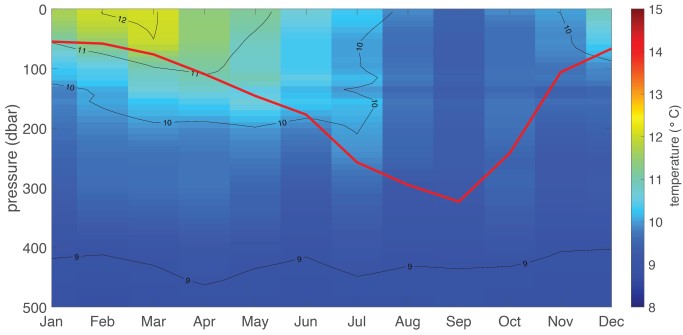

**Figure 6.** Climatology of upper ocean temperature based on sensor data from all SOTS moorings (SOTS, PULSE, SAZ), and associated mixed-layer depth.

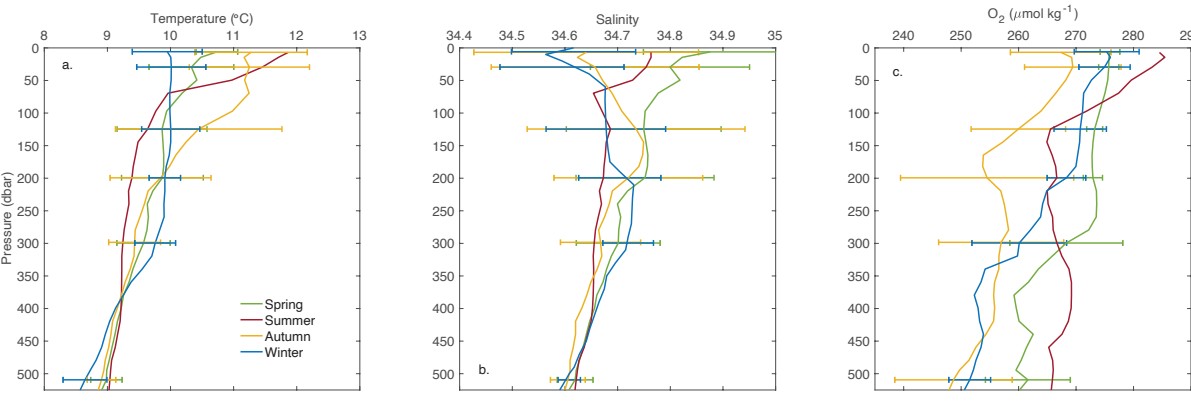

**Figure 7.** Seasonal profiles from shipboard CTD casts at the SOTS site. a) Temperature, b) Salinity, c) dissolved oxygen ($O_2$). The profiles are mean values compiled from all voyages to the site in a particular season. The horizontal bars indicate the standard deviation associated with the mean profile at a subset of depths; the summer season profile was acquired from a single voyage and so has no associated standard deviation.

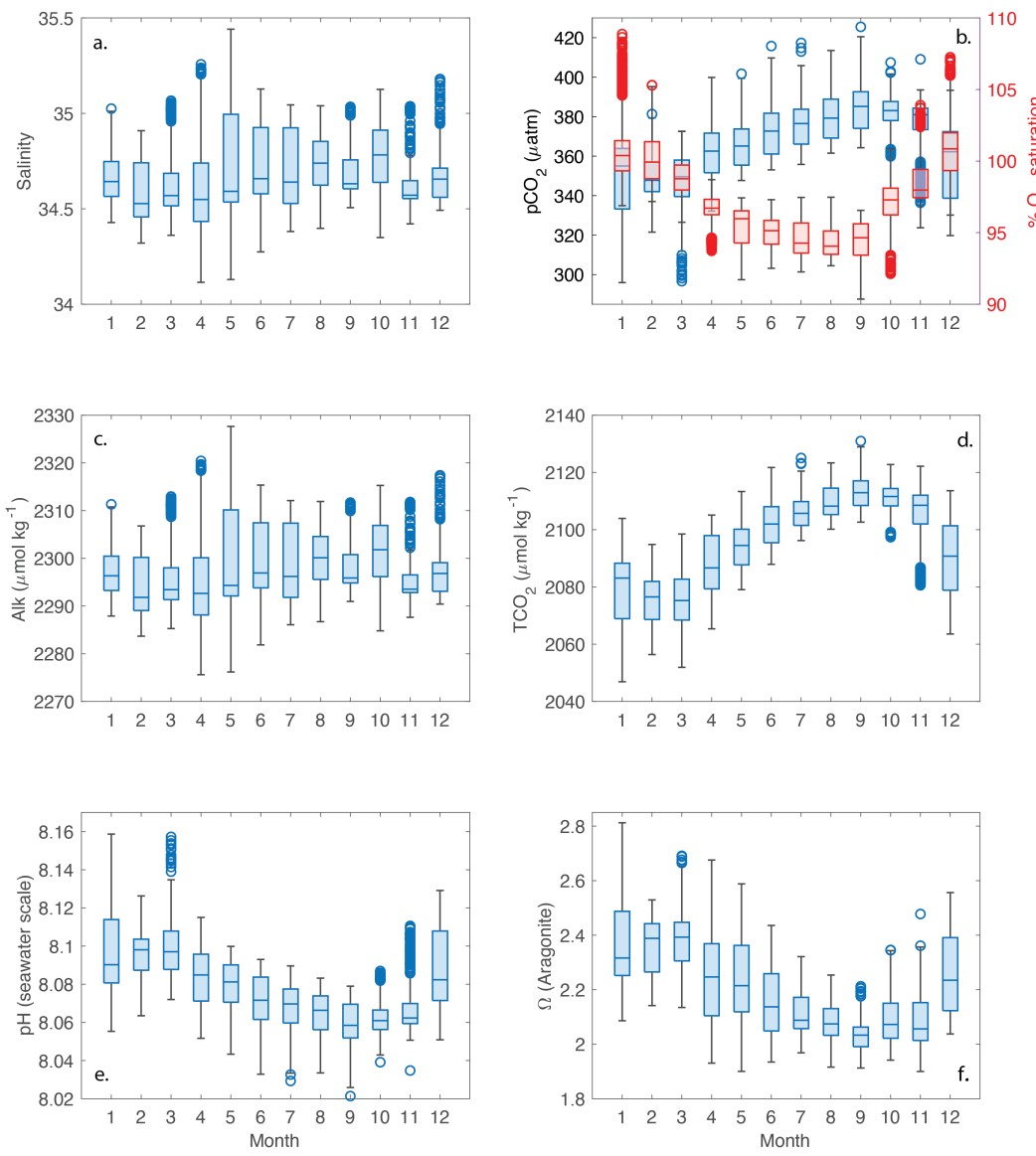

**Figure 8.** Climatological seasonal cycles of surface (~1 m depth) water hydrography and carbonate system parameters based on sensor data from the SOFS moorings between 2011 and 2021: a) salinity; b) $CO_2$ partial pressure ($pCO_2$) and oxygen saturation; c) alkalinity; d) dissolved inorganic carbon ($TCO_2$), e) pH, and f) aragonite saturation state ($\Omega$). Note that the data shown in panels c - f were computed based on the salinity and $pCO_2$ observations in panels a and b.

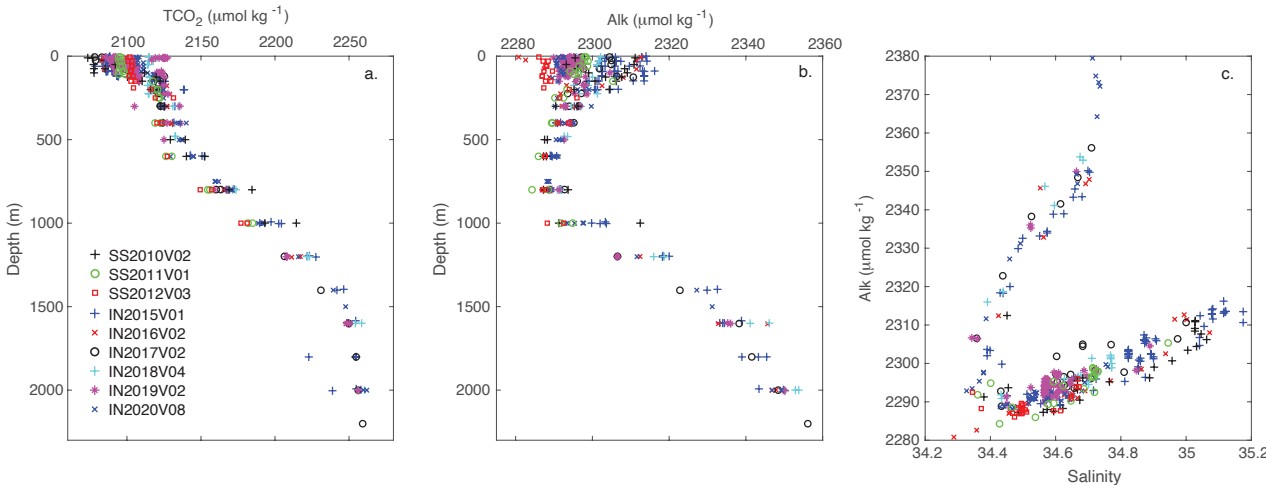

**Figure 9.** The $CO_2$ system at the SOTS site. a) dissolved inorganic carbon ($TCO_2$) as a function of depth, b) total alkalinity (Alk) as a function of depth, and c) the relationship between alkalinity and salinity. The linear relationship between Alk and salinity in the upper ocean (0 -800 m, lower values of Alk over a wide range of salinity) can be described by the equation TA = 36.6S + 1027 (n = 360, $R^2$ = 0.86); the erorr associated with the fit is $\sim$7 $\mu$mol kg$^{-1}$). Observations shown in all three panels were collected between 2010 and 2020 (see also Fig. 4).

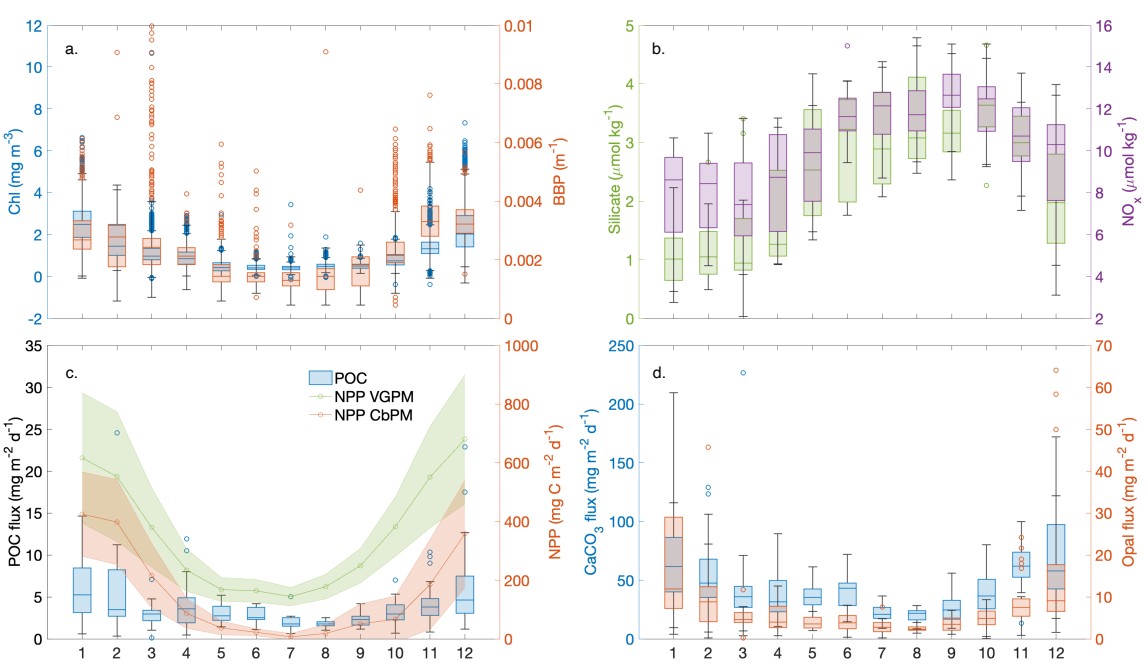

**Figure 10.** Climatological seasonal cycles of nutrients and biomass proxies as well as primary productivity and the export of organic carbon. a) chlorophyll (left-hand axis and in blue) and particulate backscatter (right-hand axis and in orange) from sensors at 30m depth on the SOFS mooring; b) silicate (left-hand axis and in green) and nitrate (right-hand axis and in purple) from discrete samples collected roughly fortnightly on the SOFS mooring; c) particulate organic carbon (POC; left-hand axis and in blue) flux collected by a sediment trap deployed at 2000m on the SAZ mooring and based on observations collected between 1997 and 2021, and net primary production (NPP) from two satellite-based models (right-hand axis and in green, VGPM, and orange, CbPM); and d) the composition of particulate fluxes at 2000m with $CaCO_3$ in blue (left-hand axis) and opal (biogenic silica, right-hand axis) in orange.

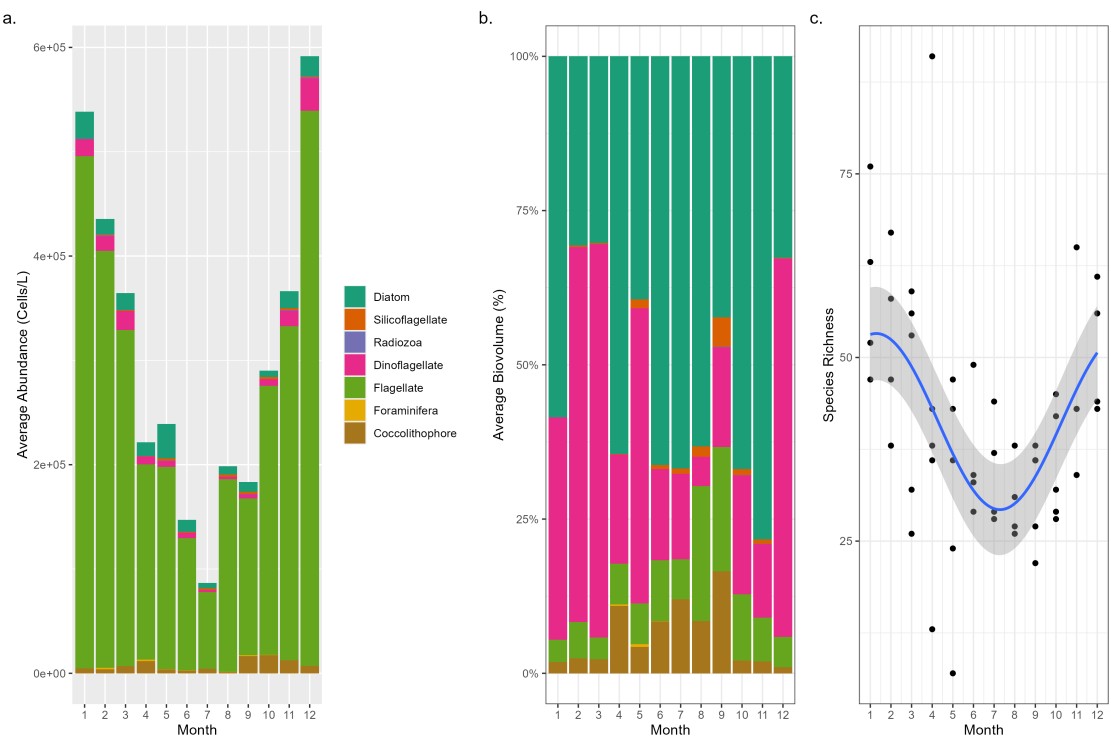

**Figure 11.** Climatological seasonality of phytoplankton community composition at the STOS site. a) abundance; b) biovolume; c) species richness. The colours in panels a and b indicate the dominant taxa.

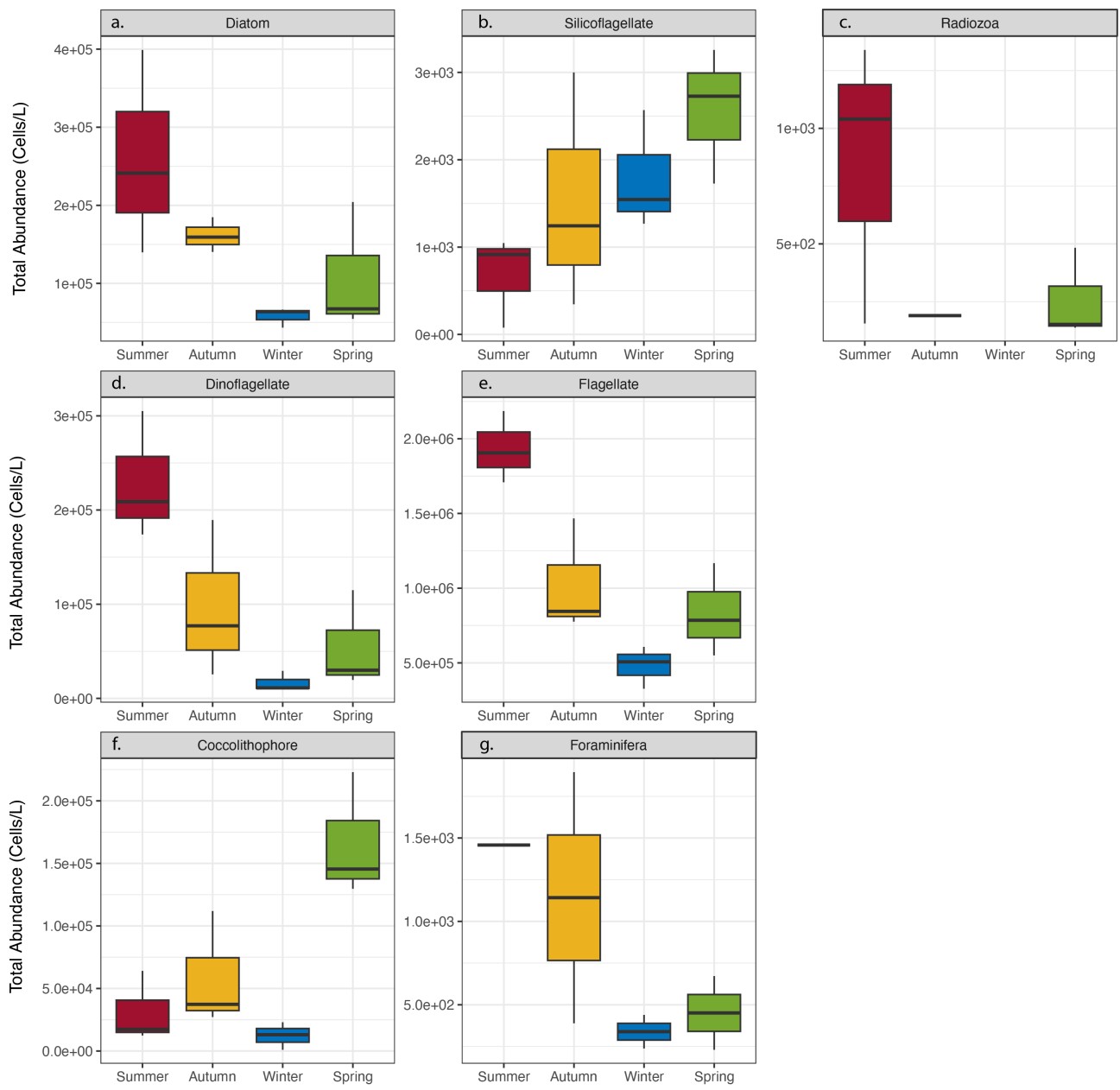

**Figure 12.** Climatological seasonal cycles of the dominant micro-plankton taxa at the SOTS site. Top row: silicifiers including a) diatoms, b) silicoflagellates and c) radiozoa; middle row: flagellate taxa including d) dinoflagellates and e) flagellates; and bottom row: calcifying taxa including f) coccolithophores and g) foraminifera.

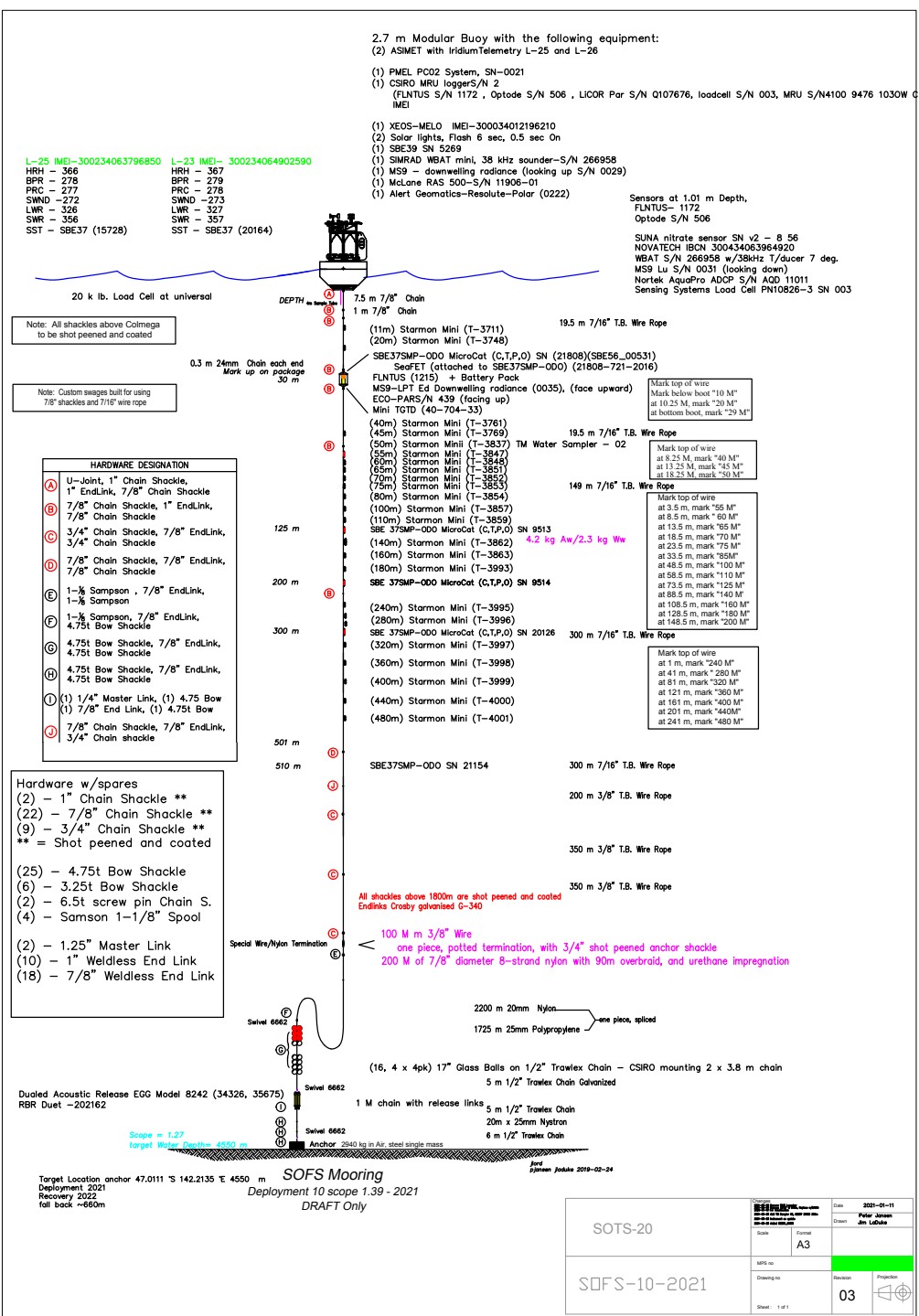

**Figure A1.** The SOFS mooring. An example of the mooring drawing used for the 2021 (SOFS-10) construction and deployment. The SOFS mooring design was provided by the Woods Hole Oceanographic Institution. The drawing is best viewed printed on large format paper; the online version is high-resolution to allow the small font sizes to be read.

| Mooring | Deployment | Recovery | Latitude | Longitude |
|---|---|---|---|---|
| SAZ47-01-1997 | 26-Sep-1997 | 17-Feb-1998 | -46.759 | 142.070E |
| SAZ47-02-1998 | 27-Mar-1998 | 05-Feb-2000 | -46.759 | 142.090E |
| SAZ47-03-1999 | 31-Jul-1999 | 30-Jul-2000 | -46.759 | 142.099E |
| SAZ47-04-2000 | 13-Oct-2000 | 13-Oct-2001 | -46.783 | 142.039E |
| SAZ47-07-2003 | 28-Sep-2003 | 26-Sep-2004 | -46.908 | 141.646E |
| SAZ47-09-2005 | 17-Dec-2005 | 12-Oct-2006 | -46.800 | 141.844E |
| SAZ47-11-2008 | 04-Oct-2008 | 22-Sep-2009 | -44.792 | 145.639E |
| Pulse-6-2009 | 28-Sep-2009 | 18-Mar-2010 | -46.322 | 140.678E |
| SAZ47-12-2009 | 29-Sep-2009 | 09-Sep-2010 | -46.834 | 141.657E |
| SOFS-1-2010 | 18-Mar-2010 | 20-Apr-2011 | -46.723 | 141.954E |
| Pulse-7-2010 | 12-Sep-2010 | 17-Apr-2011 | -46.935 | 142.258E |
| SAZ47-13-2010 | 12-Sep-2010 | 04-Aug-2011 | -46.830 | 141.650 |
| Pulse-8-2011 | 04-Aug-2011 | 19-Jul-2012 | -46.930 | 142.215E |
| SAZ47-14-2011 | 04-Aug-2011 | 21-Jul-2012 | -46.794 | 141.816E |
| SOFS-2-2011 | 25-Nov-2011 | 22-Jul-2012 | -46.772 | 141.987E |
| SOFS-3-2012 | 14-Jul-2012 | 01-Jan-2013 | -46.664 | 142.063E |
| Pulse-9-2012 | 17-Jul-2012 | 05-May-2013 | -46.849 | 142.399E |
| SAZ47-15-2012 | 18-Jul-2012 | 08-Oct-2013 | -46.837 | 141.679E |
| SOFS-4-2013 | 01-May-2013 | 14-Oct-2013 | -46.777 | 141.993E |
| SAZ47-16-2013 | 04-May-2013 | 26-Mar-2015 | -46.793 | 141.823E |
| Pulse-10-2013 | 07-May-2013 | 13-Oct-2013 | -46.938 | 142.285E |
| SOFS-5-2015 | 24-Mar-2015 | 13-Apr-2016 | -46.667 | 142.073E |
| Pulse-11-2015 | 25-Mar-2015 | 19-Mar-2016 | -46.938 | 142.326E |
| SAZ47-17-2015 | 27-Mar-2015 | 18-Mar-2016 | -46.825 | 141.656E |
| SAZ47-18-2016 | 18-Mar-2016 | 21-Mar-2017 | -46.784 | 141.842E |
| SOFS-6-2017 | 19-Mar-2017 | 09-Nov-2017 | -46.027 | 142.129E |
| SAZ47-19-2017 | 21-Mar-2017 | 09-Mar-2018 | -46.109 | 142.308E |
| SOFS-7-2018 | 06-Mar-2018 | 16-Mar-2018 | -47.011 | 142.214E |
| SAZ47-20-2018 | 09-Mar-2018 | 21-Mar-2019 | -46.792 | 141.794E |
| SOFS-7.5-2018 | 22-Aug-2018 | 22-Mar-2019 | -47.023 | 142.233E |
| SOFS-8-2019 | 18-Mar-2019 | 04-Sep-2020 | -46.893 | 142.345E |
| SAZ47-21-2019 | 20-Mar-2019 | 03-Sep-2020 | -46.826 | 141.648E |

| SOFS-9-2020 | 01-Sep-2020 | 23-Apr-2021 | -46.985 | 141.812E |
|---|---|---|---|---|
| SAZ47-22-2020 | 02-Sep-2020 | 24-Apr-2021 | -46.794 | 141.816E |
| SOFS-10-2021 | 20-Apr-2021 | 12-May-2022 | -46.998 | 142.285E |
| SAZ47-23-2021 | 24-Apr-2021 | 11-May-2022 | -46.826 | 141.654E |

Table A1: Southern Ocean Time Series (SOTS) mooring deployment and recovery dates and locations. Note that the SOFS-6 mooring broke free of its anchor and began to drift on 01-Nov-2017 until it was recovered on 09-Nov-2017; observations collected during this period of drift have not been included in our analyses.

| Month | Depth | POC | PIC | BSi | PC | Total Mass | NPP VGPM | NPP CbPM |
|---|---|---|---|---|---|---|---|---|
| Jan | 1000 | 6.2 ($\pm$4.0) | 4.2 ($\pm$2.5) | 3.8 ($\pm$4.3) | 10.8 ($\pm$5.9) | 60.8($\pm$ 41.0) | 616.5 ($\pm$222.3) | 425.2 ($\pm$143.9) |
| | 2000 | 6.2 ($\pm$3.5) | 7.8 ($\pm$3.1) | 8.8 ($\pm$6.7) | 14.2 ($\pm$6.4) | 101.1 ($\pm$60.0) | | |
| | 3800 | 5.0 ($\pm$2.6) | 7.5 ($\pm$2.9) | 7.6 ($\pm$4.4) | 12.4 ($\pm$5.1) | 99.7 ($\pm$38.9) | | |
| Feb | 1000 | 5.2 ($\pm$5.6) | 3.8 ($\pm$3.4) | 2.7 ($\pm$2.8) | 8.7 ($\pm$8.9) | 47.2 ($\pm$50.2) | 552.9 ($\pm$220.9) | 398.0 ($\pm$144.6) |
| | 2000 | 5.6 ($\pm$5.0) | 6.5 ($\pm$3.9) | 4.8 ($\pm$4.5) | 12.1 ($\pm$8.4) | 78.1 ($\pm$57.2) | | |
| | 3800 | 5.1 ($\pm$3.6) | 7.5 ($\pm$2.8) | 7.3 ($\pm$5.4) | 12.6 ($\pm$5.9) | 98.5 ($\pm$43.5) | | |
| Mar | 1000 | 3.3 ($\pm$2.1) | 3.0 ($\pm$2.3) | 1.6 ($\pm$1.3) | 6.4 ($\pm$4.1) | 36.6 ($\pm$27.8) | 379.1 ($\pm$134.6) | 214.9 ($\pm$109.3) |
| | 2000 | 2.8 ($\pm$1.4) | 5.1 ($\pm$4.9) | 2.3 ($\pm$1.1) | 7.0 ($\pm$2.7) | 44.9 ($\pm$24.5) | | |
| | 3800 | 2.8 ($\pm$1.4) | 4.9 ($\pm$1.4) | 3.0 ($\pm$1.1) | 7.7 ($\pm$2.5) | 58.3 ($\pm$17.0) | | |
| Apr | 1000 | 3.9 ($\pm$1.9) | 2.7 ($\pm$1.3) | 1.1 ($\pm$1.2) | 6.4 ($\pm$3.4) | 34.8 ($\pm$21.2) | 234.4 ($\pm$72.1) | 86.9 ($\pm$50.8) |
| | 2000 | 4.3 ($\pm$3.1) | 4.6 ($\pm$2.5) | 2.8 ($\pm$1.8) | 8.9 ($\pm$5.5) | 59.4 ($\pm$33.3) | | |
| | 3800 | 2.6 ($\pm$1.0) | 4.8 ($\pm$1.6) | 2.9 ($\pm$1.8) | 7.2 ($\pm$2.0) | 58.1 ($\pm$19.9) | | |
| May | 1000 | 3.4 ($\pm$2.1) | 3.1 ($\pm$1.6) | 1.3 ($\pm$0.9) | 5.6 ($\pm$3.5) | 30.6 ($\pm$21.3) | 168.1 ($\pm$40.5) | 37.2 ($\pm$23.4) |
| | 2000 | 3.0 ($\pm$1.0) | 4.3 ($\pm$1.3) | 1.8 ($\pm$0.7) | 7.2 ($\pm$1.6) | 48.3 ($\pm$16.2) | | |
| | 3800 | 2.4 ($\pm$0.9) | 4.4 ($\pm$1.1) | 2.3 ($\pm$1.3) | 6.8 ($\pm$1.7) | 51.4 ($\pm$13.1) | | |
| Jun | 1000 | 3.0 ($\pm$1.2) | 3.0 ($\pm$1.7) | 1.2 ($\pm$1.0) | 6.1 ($\pm$3.1) | 36.6 ($\pm$21.8) | 163.6 ($\pm$38.6) | 22.5 ($\pm$11.4) |
| | 2000 | 2.8 ($\pm$0.9) | 4.7 ($\pm$1.8) | 1.9 ($\pm$1.1) | 7.5 ($\pm$2.5) | 52.6 ($\pm$20.1) | | |
| | 3800 | 2.7 ($\pm$1.3) | 4.1 ($\pm$1.5) | 2.6 ($\pm$1.7) | 6.8 ($\pm$2.5) | 50.0 ($\pm$18.2) | | |
| Jul | 1000 | 2.3 ($\pm$1.4) | 1.9 ($\pm$1.4) | 1.0 ($\pm$0.8) | 3.9 ($\pm$2.6) | 22.8 ($\pm$16.4) | 143.9 ($\pm$31.1) | 6.8 ($\pm$11.5) |
| | 2000 | 2.1 ($\pm$1.1) | 2.7 ($\pm$1.0) | 1.4 ($\pm$0.9) | 4.8 ($\pm$1.7) | 32.6 ($\pm$13.7) | | |
| | 3800 | 1.7 ($\pm$0.5) | 3.4 ($\pm$1.0) | 1.9 ($\pm$1.1) | 5.1 ($\pm$1.4) | 39.7 ($\pm$11.8) | | |
| Aug | 1000 | 1.9 ($\pm$1.0) | 1.5 ($\pm$0.8) | 0.7 ($\pm$0.5) | 3.5 ($\pm$2.0) | 21.2 ($\pm$11.9) | 177.8 ($\pm$41.6) | 18.5 ($\pm$30.4) |
| | 2000 | 1.8 ($\pm$0.4) | 2.4 ($\pm$0.7) | 1.2 ($\pm$0.3) | 4.2 ($\pm$1.0) | 29.6 ($\pm$7.9) | | |

| | | POC | PIC | BSi | PC | Total mass | VGPM | CbPM |
|---|---|---|---|---|---|---|---|---|
| | 3800 | 1.7 (±0.6) | 2.7 (±0.5) | 1.4 (±0.6) | 4.4 (±0.8) | 32.6 (±5.9) | | |
| Sep | 1000 | 2.9(±1.5) | 2.5 (±1.5) | 1.4 (±0.8) | 5.0 (±2.8) | 29.4 (±16.7) | 249.6 (±58.7) | 51.3 (±67.8) |
| | 2000 | 2.3 (±0.8) | 3.1 (±1.3) | 1.7 (±0.8) | 5.1 (±1.8) | 33.2 (±15.9) | | |
| | 3800 | 1.3 (±0.4) | 2.7 (±0.8) | 1.4 (±0.5) | 4.1 (±1.1) | 31.5 (±9.4) | | |
| Oct | 1000 | 4.4 (±2.3) | 5.8 (±3.4) | 2.7 (±1.8) | 10.2 (±5.7) | 68.7 (±40.8) | 382.1 (±101.1) | 66.7 (±78.9) |
| | 2000 | 3.3 (±1.4) | 4.7 (±2.2) | 2.4 (±1.0) | 7.9 (±3.1) | 53.8 (±27.0) | | |
| | 3800 | 1.9 (±0.9) | 4.0 (±1.3) | 1.9 (±0.7) | 6.0 (±2.2) | 46.3 (±15.2) | | |
| Nov | 1000 | 4.2 (±1.9) | 5.6 (±2.7) | 2.3 (±1.7) | 9.6 (±4.6) | 60.6 (±36.5) | 551.4(±172.7) | 185.5 (±152.7) |
| | 2000 | 4.3 (±2.1) | 7.6 (±2.4) | 4.1 (±2.5) | 11.9 (±4.2) | 86.8 (±34.1) | | |
| | 3800 | 3.2 (±1.9) | 6.3 (±2.3) | 2.6 (±0.8) | 9.4 (±4.1) | 71.2 (±25.6) | | |
| Dec | 1000 | 4.4 (±3.8) | 3.5 (±2.7) | 4.3 (±4.7) | 7.3 (±6.0) | 44.1 (±42.8) | 680.1 (±219.6) | 357.6 (±182.9) |
| | 2000 | 6.6 (±5.1) | 8.3 (±4.4) | 7.8 (±7.9) | 15.0 (±8.8) | 104.3 (±68.9) | | |
| | 3800 | 2.9 (±1.9) | 5.5 (±1.7) | 3.2 (±2.3) | 8.4 (±3.3) | 65.1 (±24.0) | | |

Table A2: Montly mean (standard deviation) values of export and particle composition at the SOTS site from sediment traps deployed at 1000m, 2000m and 3800 m between 1997 and 2021. The POC, PIC, BSi, PC, and Total mass are given in units of (mg m$^{-2}$ d$^{-1}$). Also shown are two satellite-based estimates of NPP: VGPM from 2012 - 2022 and CbPM from 2012 - 2020, both given in untis of (mg C m$^{-2}$ d$^{-1}$).

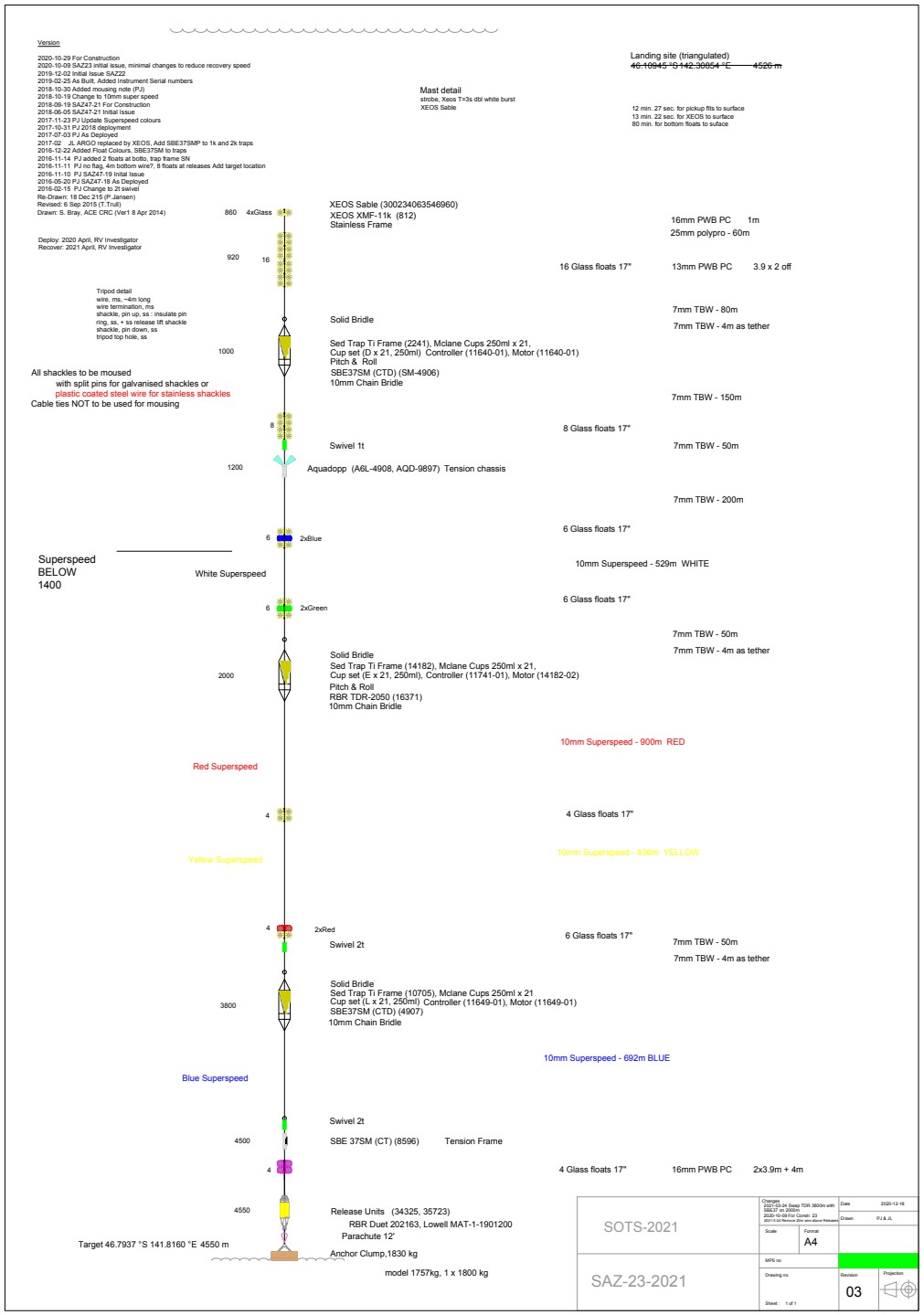

**Figure A2.** The SAZ mooring. A example of the mooring drawing used for the 2021 (SAZ-23) construction and deployment. The drawing is best viewed printed on large format paper; the online version is high-resolution to allow the small font sizes to be read.

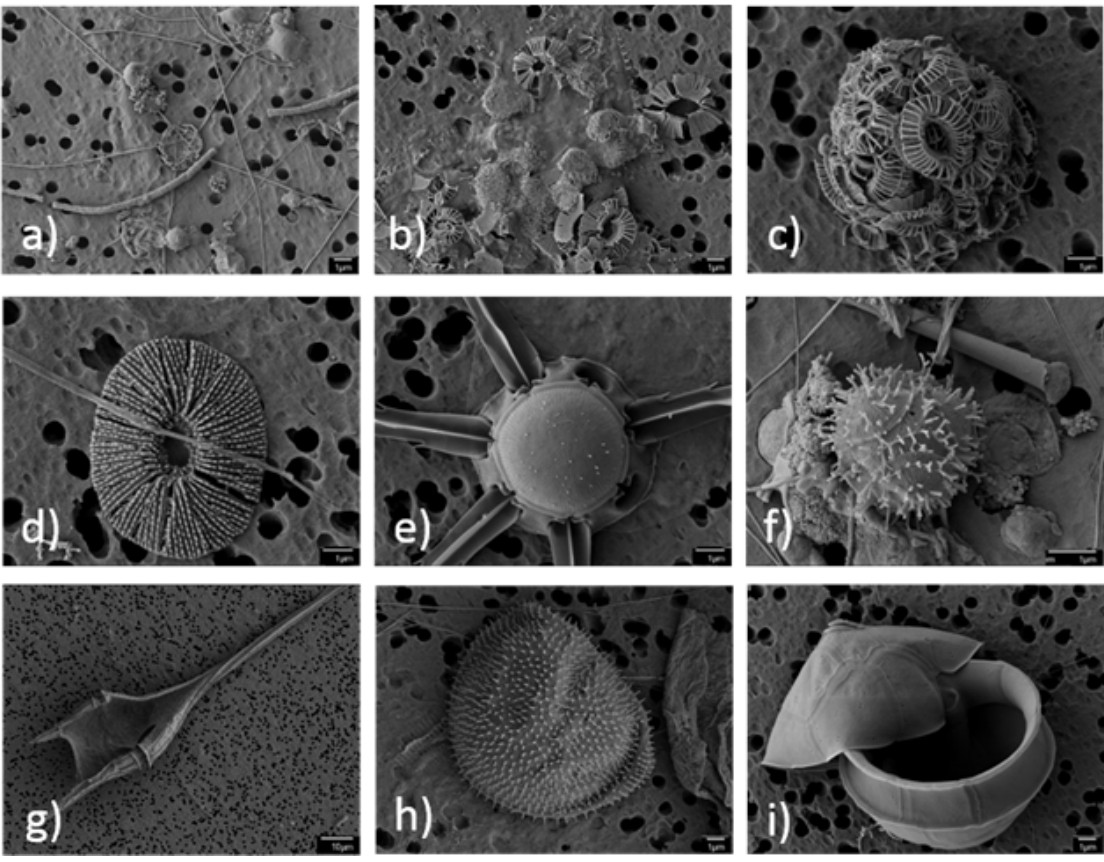

**Figure A3.** Scanning Electron Microscopy (SEM) images showing a range of taxonomic groups. a) flagellate form of *Phaeocystis antarctica* with characterisitic star array; b) collapsed coccolithophorid cell including both heterococcolith and halococccoliths; c) calcification morphotype B/C for *Emiliania huxleyi*; d) individual lith of calcifying species *Umbellosphaera tenuis*; e) siliceous *Corethron pennatum* (diatom); f) siliceous *Triparma strigata* (parmales); g) dinoflagellate *Tripos lineatus*; h) dinoflagellate *Prorocentrum* cf *compressum*; and i) dinoflagellate *Scripsiella trochidea*.