# Peer review of "The Southern Ocean Time Series: A climatological view of hydrography, biogeochemistry, phytoplankton community composition, and carbon export in the Subantarctic Zone"

_EGUsphere, 2024_

## Referee Comment (RC2)

**Review of Shadwick et al. "The Southern Ocean Time Series: A climatological view of hydrography, biogeochemistry, phytoplankton community composition, and carbon export in the Subantarctic Zone" submitted to EGUsphere (manuscript #egusphere-2024-3887)**

The paper describes the seasonal climatology of several monitored ocean parameters (physics, carbonate system, nutrients, phytoplankton and export flux) based on the decades-long time-series maintained at the SOTS (Southern Ocean Time Series) site in the Australian sector of the circum-global Subantarctic Zone (SAZ).

The paper is well-written, informative and provides a broader context for the myriad of papers that have been written about the SOTS site over many years on a range of oceanographic topics, including air-sea flux, trace metal chemistry and zooplankton. It provides a summary of mean climatologies for the selected parameters while also highlighting the associated knowledge gained from these other focussed studies.

One comment is that the paper needs to highlight that novelty of the climatological approach within the context of the wealth of other data generated by the time-series programme, which is almost the focus of the Discussion. Furthermore, the paper would benefit from analysing the interpretations of the SOTS data as presented with respect to other HNLC regions, such as the subarctic North Pacific and other regions of the SAZ. This would require an additional section to be added to the Discussion section.

It was good to see the efforts of the SOTS stalwart, Tom Trull, recognised in the Acknowledgements as the persistence of certain people is often the gel that enables such valuable datasets to be generated over the required timeframes to account for the inherent variability of ocean parameters over a variety of space and temporal scales.

Except for the recommendation of an additional section in the Discussion to place the SOTS data in the context of other HNLC regions, this review only has minor comments on the manuscript as detailed below:

(1) In several places, "time series" should be hyphenated. Similarly, "mixed-layer depth". The authors should ensure consistency throughout the text.

(2) Lines 69-70: how were the RAS samples preserved for the different parameters that were measured? Were the phytoplankton identifications, in particular, affected by the use of mercuric chloride as intimated in the caption for Figure A4?

(3) Lines 74-75: it is not clear if the instrumentation measuring "subsurface temperature, salinity and dissolved oxygen" were at different depths to measure surface mixed layer processes or if the measurements were just made at static (surface, 30m or 50 m depths) as for some of the other instrumentation on the SOFS mooring.

(4) Lines 78-79: "McLane" has a capital "L". What model of PARFLUX trap was used on the SAZ mooring over the years? Did it change? How were the sequential sediment trap samples preserved?

(5) Lines 95-96: a definition is provided for the STF. Should one also be provided for the SAF? Is there any variability in the location of the SAF that could affect the surface ocean parameters at the SOTS site?

(6) Lin 123: "$p$CO$_2$" should have the "p" italicised. Correct in many other places in the text.

(7) Line 127: how "far north" of the SOTS site was deemed "to far" too be included in the climatologies?

(8) Line 133: shouldn't "PULSE" be capitalised?

(9) Lines 135 and 146-147: Looking at Figure 5b there seems to be some correlation between deep mixed-layer depth formation in autumn and winter and the southernmost position of the STF, especially within the small SOTS box. Yet, in the last lines (148-149) in the paragraph, the sentence states that deep mixed-layer depths autumn and winter are related to the STF being further north. Please comment as this doesn't appear to be the pattern shown in Fig, 5b (unless the scales are precluding close enough scrutiny).

(10) Lines 150-151: "summer maxima of 15°C ..... (Fig. 5b)". This is not apparent in the climatology presented in Fig. 6, presumably due to variability – this figure suggests the summer climatological maxima at the site is only 12-13°C. Please comment further on these observations.

(11) Line 154: "Subtropical" should be capitalised as for "Subantarctic" later in the sentence.

(12) Line 161-179 "Deep-ocean observations at the SOTS site": It is unclear of the relevance of this section as the implications of these deep-water observations are not discussed further in the Discussion.

(13) Lines 203-204: over what depths does "deep (Alk-rich) waters" refer to? Similarly, "shallow depths" later on in the sentence.

(14) Line 205: As mentioned previously, "Subtropical" should be capitalised, plus perhaps amended to "Subtropical surface waters" in the context discussed here.

(15) Line 214: last part of sentence should read: "... through to June (winter) in most years". Adding "(winter)" will assist Northern Hemisphere readers to orient themselves to the Southern Hemisphere seasonality.

(16) Lines 216-218: Is a reference needed at the end of the sentence here where "regenerated nutrients" are discussed when these are not reported on by the paper.

(17) Line 220: spell out "foraminifera", rather than "forams" as for the other groups of organisms.

(18) Line 231: add semi-colon (";") between "... SOTS site" and "however", rather than a comma (",").

(19) Lines 232-234: isn't the apparent non-seasonality in silicoflagellate abundance (Fig. 13) also warranted some discussion here?

(20) Lines 249-251: why are no values provided for the flux "transfer efficiency".

(21) Lines 254-257: Long sentence – consider re-writing.

(22) Line 261: "... interannual variability" of what specifically?

(23) Line 261-262: "do you mean "... multiple linear regression analysis and ..." ?

(24) Line 266: add "(NCP" after "Net community production".

(25) Line 272: "The seasonal succession away from a phytoplankton-dominated community" – not sure what this actually means and how this can be "elucidated on the basis of chlorophyll and backscatter data" – please comment.

(26) Line 273: "ship board" is written as "shipboard" (preferred) and "ship-board" in other places in the text. Similarly, "time scales" (e.g., line 324). Please ensure consistency.

(27) Lines 276-275: no PAR data are shown in the paper yet Fe and Mn are inferred as seasonally co-limiting. Is light also a factor, especially in winter with such deep mixed layers?

(28) Line 279: "cocolithophores" is missing one of the "c"s.

(29) Line 280: "there is a seasonal depletion" – when?

(30) Line 281: "silica" should be "silicate" as it refers to the macronutrient, not the mineral.

(31)   Line 281: "similar to the global median" – what is this value? Should be stated here for ease of reference.

(32)   Line 283: add "carbon" between "biological" and "pump".

(33)   Line 301: "an independent measure of particle export" – was this at the SOTS site or in the SAZ in general?

(34)   Lines 305-316: the work by Wilks et al. on diatom fluxes and species identifications at SOTS is not references here.

   (a) Wilks, J. V., et al. (2017). "Biogeochemical flux and phytoplankton succession: A year-long sediment trap record in the Australian sector of the Subantarctic Zone." Deep Sea Research Part I: Oceanographic Research Papers **121**: 143-159.

   (b) Wilks, J. V. and L. K. Armand (2017). "Diversity and taxonomic identification of *Shionodiscus* spp. in the Australian sector of the Subantarctic Zone." Diatom Research **32**(3): 295-307.

(35)   Line 306: not sure what "modern and historical properties of diatoms" means.

(36)   Line 310: need space between "*huxleyi*" and "is".

(37)   Lines 325-337 "Links to Higher Trophic Levels": how do these studies relate back to the climatology work presented in the paper? As outlined here, it appears that this discussion is just outlining other studies without reflecting on the relationships to the new research that is the core aspect of the paper.

(38)   Line 334: do you mean "particulate organic carbon"? Also, "low surface (chlorophyll or plankton) biomass" later in the same sentence?

(39)   Lines 335-337: the last sentence seems to be somewhat tacked on but does emphasise the other co-benefits of sediment trap records with respect to plankton species identifications.

(40)   Line354: "... future impacts." – of what? On what?

(41)   Line 364: "... RV *Investigator*" should be italicised.

(42)   Line 420: species name should be italicised.

(43)   Line 425: missing the opening speech bracket from the title.

(44)   Line 489: extra comma after "... A. S.,". Similarly on line 493.

(45)   Lines 490, 498 and 501: species names should be italicised.

(46)   Lines 513-517: do all the authors need to be listed here (Sathyendranath et al.)? Maybe list first 10 and then add "et al." Similarly, lines 549-552 (Bates et al.) and lines 554-557 (Takahashi et al.).

(47)   Figure 1: could the mean circulation patterns and main surface water masses be also shown on this figure, especially for international readers not familiar with this region?

(48)   Figure 2: by showing the climatologies of the various fronts on this figure raises questions about how variable the SAF and PF are? In particular, do shifts in the location of the SAF affect the SOTS site?

(49)   Figure 3: does "sensor" mean shipboard "CTD" or moored "CTD" data or something else? Are these data from the SOTS site only rather than from other deployments in the wider SAZ? State this in the caption.

(50)   Figures 5b and 6: what is the definition that is used to define the "mixed-layer depth" calculations?

(51)   Figure 7: is there any way of clearly showing the variability associated with the CTD data?

(52)   Figure 8b: Why are there no data from 3900 m shown?

(53)   Figure 8c: the pressure differences shown here suggest the sensors were moored at slightly different depths. So, did this affect other moored data collected at the same

time or were the moorings adjusted to account for the slight depth variations between mooring deployments?

(54)        Figure 9: Axis labels are too small to read. Consider adding labels to each of the plots for ease of clarity.

(55)        Figure 9: are all of these climatologies generated from within the mixed layer or from a specific water depth?

(56)        Figure 10: Axis labels are too small to read. Consider adding labels to each of the plots for ease of clarity. Can $r^2$ values be ascribed to the linear relationships in the Alk v S plot?

(57)        Figure 11: Axis labels are too small to read. Consider adding labels to each of the plots for ease of clarity. What production models do the green and orange plots, respectively, in Figure 11c refer to?

(58)        Figure 12: the colour scale on these plots, especially 12b, make it very difficult to see the difference sin phytoplankton community composition as described. Please adjust.

(59)        Table A1: add "°S" and "°E" to the last two table columns, respectively.

(60)        Figures A1 and A2: nice to see these detailed mooring schematics but the small font sizes make it difficult/impossible to read any of the details.

(61)        Figure A3: What water depths were these T and S climatologies from? In Figure A3b, what are the bottom red dashes – extreme outliers or artefacts?

(62)        Figure A4: what water depth were these SEM samples collected from? Presume they were collected using the RAS, which needs to be specified here, presuming they are from the SOTS site and not elsewhere in the SAZ? What was the effect of the mercuric chloride preservation on the ease of phytoplankton IDs?

---

## Author Response (AR1)

We thank the reviewer for their constructive feedback. Below we have provided (in plain text) a response to each comment (included in red).

The authors present an updated (?) view of the SOTS program. The manuscript is written as a 'summary' of previously published works at this station. As detailed below there is little/no new meaningful analysis.

Results seems to have too much reference to other work. In fact it is difficult to see what this manuscript is contributing new to science. It appears that many of the results have been presented in other publications already. I am all for papers from time-series sites resummarizing results, but there does need to be something new (new data, new trend, new interpretation, etc.).

Perhaps this manuscript provides updated time-series, but it needs to be more clearly presented what is new.

Thank you for this helpful perspective. This manuscript brings together climatological seasonal cycles of the physical, chemical, and biological observations collected at the SOTS site over many, many years. Importantly, this is the first time the observations have been presented, and interpreted together, and the first time that we have achieved full seasonal resolution for some parameters. For example, this is the first full annual record of both the calcifying and non-calcifying phytoplankton communities. We have revised the Introduction (lines 35-49 in the revised text) of the manuscript to better highlight the novelty of this work. We have also included a new section evaluating trends in several parameters (section 3.5 in the revised text; see also response below).

For example, Figure 8, why isn't there more discussion of the temperature and salinity records. Are changes here real or is this a data quality/standardization issue that needs to be resolved?

Thank you for the comment; in response to feedback from an additional reviewer, we have removed the section on deep ocean temperature and salinity from the revised version. We have also expanded the description of the seasonality in temperature and added detail on the differences between the gridded monthly ocean temperature product in the upper 500 m and the seasonal climatology (lines 162-165 in the revised text).

FIgures 9, 10, why aren't the CO2 system parameters shown in a time-series?
Is there ocean acidification going on like in other ocean regions?

Thank you for the questions. The CO2 system parameters, like the physical and biological properties are shown as seasonal climatologies, as this is the focus of the manuscript. There is indeed a signal of ocean acidification in the observations collected at the SOTS site, and this is the focus of a recent publication (Shadwick et al., 2023, which includes the pCO2 time series), and therefore not emphasised here. We have included an explicit reference to the long-term change in surface pCO2 in a new results section in the revised manuscript (section 3.5 in the revised text).

We have also added a new Results section that evaluates long term trends in the parameters for which we have sufficiently long records (temperature, salinity, pCO2,

nitrate, silicate, NPP, as well as POC, PIC, BSi at 2000m). Interestingly, apart from the pCO2, which exhibits a trend attributed to ocean acidification, none of the parameters indicate statistically significant changes in time, even those with records exceeding 26 years.

Figure 12c - I didn't see a specific discussion/analysis of species richness?
Thank you for the comment; we have included a description of the species richness (see text below; lines 255-258 in the revised text) in the revised manuscript.

"The seasonality of species richness at the SOTS site (Fig. 11c) closely follows total abundance (Fig. 11a), chlorophyll (Fig. 10a) and NPP (Fig. 10c), with minima in winter and highest values observed during the productive season, or (austral) summer months. The total number of taxa recorded at the SOTS site since the inception of taxonomic sampling is over 300 and is a result of the combination of both SEM and LM examination."

There are few, if any sustained observations of the phytoplankton and microzooplankton community in the Subantarctic Zone for comparison (see Gutiérrez-Rodríguez et al, 2022), highlighting the importance of the SOTS program in resolving diversity and abundance through microscopy approaches. There is insufficient sample returned by the RAS to resolve diversity through molecular techniques, using currently available methods.

Andres Gutiérrez-Rodríguez, Adriana Lopes dos Santos, Karl Safi, Ian Probert, Fabrice Not, Denise Fernández, Priscillia Gourvil, Jaret Bilewitch, Debbie Hulston, Matt Pinkerton, Scott D. Nodder,Planktonic protist diversity across contrasting Subtropical and Subantarctic waters of the southwest Pacific, Progress in Oceanography, Volume 206, 2022, 102809, ISSN 0079-6611, https://doi.org/10.1016/j.pocean.2022.102809.

At a minimum, suggest a combined section if allowed by the journal.
Thank you for the suggestion. We have revised the Introduction and Results section to emphasise the novelty of the seasonal climatologies presented and added a new section to the Results evaluating long term Trends. We have elected to keep the Results and Discussion as individual sections in the manuscript.

Section 3.4.  It is confusing the jumping back and forth between abundance and biovolume. L225-230 there is mention that dynamics in the phytoplankton community are 'likely' related to variability in aspects of the physics - has no formal analysis been done to determine if this is indeed the case?
We have restructured section 3.4 as suggested (see also the response above about additional detail for species richness), in particular clarifying that seasonal evolution of the phytoplankton community has been linked to variability in the mixed-layer. The formal analysis that the reviewer points to is indeed the focus of another manuscript, and goes beyond the scope of the climatological seasonal investigation reported here.

The seasonality appears to be discussed for each major group separately rather than holistically. This section should be tightened up and more robust analyses conducted.
This section has been revised for clarity as suggested.

L290-295. Perhaps discuss why the genetic variability may mitigate the notion that acidification will decrease coccolithophores.
Thank you for the comment; we have clarified that the work indicates that ocean acidification may not mean a shift away from heavily-calcified coccolithophores as genetic variability seems to influence the response of E. hux morphotypes to changes in ocean carbon chemistry.

L310-315. Discuss what the differences are between E.hux and other coccolithophores that lead to 'conclusion' (presented in a previous paper) that Ehux is not the dominant source of exported carbonate.
Thank you for the comment. Although *E. hux* is typically the most abundant coccolithophore taxa present in samples collected at the SOTS site, it has lower biomass than larger taxa *Calcidiscus leptocylindrus* or *Helicosphaera carteri*, and so makes a smaller contribution to the export of particulate inorganic carbon (carbonate).

We thank the reviewer for their positive and constructive feedback. Below we have provided (in plain text) a response to each comment (included in red).

One comment is that the paper needs to highlight that novelty of the climatological approach within the context of the wealth of other data generated by the time-series programme, which is almost the focus of the Discussion.
Thank you for the suggestion, we have revised the Introduction, and some of the Results sections to highlight the novelty of the climatological seasonal cycles presented in this paper, many for the first time.

Furthermore, the paper would benefit from analysing the interpretations of the SOTS data as presented with respect to other HNLC regions, such as the subarctic North Pacific and other regions of the SAZ. This would require an additional section to be added to the Discussion section.
We appreciate the suggestion and agree that the manuscript will benefit from a focused section placing our results at the SOTS site into the context of other HNLC regions, such as the subarctic North Pacific. A new discussion section has been added to the revised manuscript (section 4.) which compares seasonality at the SOTS site with that at the KNOT/K2 site (western North Pacific) and at Ocean Station Papa (eastern North Pacific).

It was good to see the efforts of the SOTS stalwart, Tom Trull, recognised in the Acknowledgements as the persistence of certain people is often the gel that enables such valuable datasets to be generated over the required timeframes to account for the inherent variability of ocean parameters over a variety of space and temporal scales.
We wholeheartedly agree and remain grateful to Tom for his efforts to initiate and maintain this time series.

Except for the recommendation of an additional section in the Discussion to place the SOTS data in the context of other HNLC regions, this review only has minor comments on the manuscript as detailed below:

In several places, "time series" should be hyphenated. Similarly, "mixed-layer depth". The authors should ensure consistency throughout the text.
Corrected

Lines 69-70: how were the RAS samples preserved for the different parameters that were measured? Were the phytoplankton identifications, in particular, affected by the use of mercuric chloride as intimated in the caption for Figure A4?
All samples in current deployments are preserved using mercuric chloride, and aliquots drawn for various analyses (nutrients, total alkalinity, phytoplankton) from each sample point (bag). Selecting suitable preservatives for year-long deployments it not simple, and mercuric chloride (also used for sediment trap samples) has been found to offer consistent preservation over time. The annual deployment/retrieval voyages offer the chance to collect samples from the CTD, at multiple depths to examine freshly preserved, and in the case of air-dried filters for coccolithophores, unpreserved samples for comparison with the RAS. We see no evidence of poor preservation and features

required for identification or phytoplankton and microzooplankton (e.g thecal plate arrangement for dinoflagellates, complete frustules for diatoms, chloroplast arrangement and fibril production for Phaeocystis, lorica structure for tintinnid ciliates, coccolith structure for coccolithophores etc.) through comparing our images to published taxonomic literature and identification resources. The combination of SEM and LM underpins our approach to quantifying diversity and abundance at the SOTS. Additionally, we have worked with colleagues who extensively examined the sediment trap material (Armand, Rigual-Hernandez) to cross compare species lists and identifications at the SOTS site.

Lines 74-75: it is not clear if the instrumentation measuring "subsurface temperature, salinity and dissolved oxygen" were at different depths to measure surface mixed layer processes or if the measurements were just made at static (surface, 30m or 50 m depths) as for some of the other instrumentation on the SOFS mooring.

The subsurface instruments are mounted at static depths (30 m, 50m, 100 m, 125m, 150 m, 200 m, 250 m, 300m and 500 m in recent years), and this has been clarified in the tex (lines 77-78 in the revised version)t, with reference to the mooring diagram that shows their locations.

Lines 78-79: "McLane" has a capital "L". What model of PARFLUX trap was used on the SAZ mooring over the years? Did it change? How were the sequential sediment trap samples preserved?

The only changes over the years were 21 cup or 13 cup PARFLUX-Mark78H at the 3800m depth. All sediment trap samples were preserved with mercuric chloride; this has been included in the methods. The PARFLUX model details have been added to the text (lines 81-82).

Lines 95-96: a definition is provided for the STF. Should one also be provided for the SAF? Is there any variability in the location of the SAF that could affect the surface ocean parameters at the SOTS site?

We have used climatological positions for both the STF and SAF for Figure 1, and have added additional detail about how the SAF is defined (enhanced gradient in sea surface height; Sokolov & Rintoul, 2007). Because the SOTS site is in the northern part of the SAZ (i.e., much closer to the STF), variability in the surface ocean is more strongly influenced by variability in the location of STF. However, earlier work (Shadwick et al., 2015) has shown that advection of waters with more polar characteristics can influence properties at the SOTS site, and this point has been included in the revised text (lines 158-160).

Line 123: "pCO2" should have the "p" italicised. Correct in many other places in the text.

For consistency with earlier work, largely with the same group of authors, from the SOTS site, we have elected to leave pCO2 as plain text.

Line 127: how "far north" of the SOTS site was deemed "to far" too be included in the climatologies?
One of the deployments of the SOFS mooring (SOFS-6), and two of the SAZ moorings (SAZ-11 and SAZ-19) were at ~46S; this has been clarified in the text (line 130-131).

Line 133: shouldn't "PULSE" be capitalised?
Corrected.

Lines 135 and 146-147: Looking at Figure 5b there seems to be some correlation between deep mixed-layer depth formation in autumn and winter and the southernmost position of the STF, especially within the small SOTS box. Yet, in the last lines (148-149) in the paragraph, the sentence states that deep mixed-layer depths autumn and winter are related to the STF being further north. Please comment as this doesn't appear to be the pattern shown in Fig, 5b (unless the scales are precluding close enough scrutiny).
Thank you, this has been corrected. The correlation with autumn deep mixed-layers and the STF is indeed when the position is further south.

Lines 150-151: "summer maxima of 15°C ..... (Fig. 5b)". This is not apparent in the climatology presented in Fig. 6, presumably due to variability – this figure suggests the summer climatological maxima at the site is only 12-13°C. Please comment further on these observations.
Thank you for the comment, we have expanded the text to explicitly address the increased variability observed in the full time-series compared to the seasonal climatology, which because of the averaging of observations results in a damped seasonality (lines 163-164).

Line 154: "Subtropical" should be capitalised as for "Subantarctic" later in the sentence.
Corrected and throughout the document

Line 161-179 "Deep-ocean observations at the SOTS site": It is unclear of the relevance of this section as the implications of these deep-water observations are not discussed further in the Discussion.
We agree that the relevance of the deep ocean observations was not well established in the manuscript, and have removed this figure, and figure A4. We continue to collect observations of deep ocean temperature and salinity, and have indicated that the data are available, but not included in the manuscript, with the hope of encouraging their uptake and use by others in the community.

Lines 203-204: over what depths does "deep (Alk-rich) waters" refer to? Similarly, "shallow depths" later on in the sentence.
The deep, more alkaline waters are deeper than 800 m, the shallow depths are between the surface and roughly 800m; this has been clarified in the revised text (lines 196-198) and the caption of Figure 9.

Line 205: As mentioned previously, "Subtropical" should be capitalised, plus perhaps amended to "Subtropical surface waters" in the context discussed here.
Corrected

Line 214: last part of sentence should read: "... through to June (winter) in most years". Adding "(winter)" will assist Northern Hemisphere readers to orient themselves to the Southern Hemisphere seasonality.
Corrected, also added "Summer" to "January and March"

Lines 216-218: Is a reference needed at the end of the sentence here where "regenerated nutrients" are discussed when these are not reported on by the paper.
Thank you, references have been added in the revised version.

Lourey, M. J., and T. W. Trull (2001), Seasonal nutrient depletion and carbon export in the Subantarctic and Polar Frontal Zones of the Southern Ocean south of Australia, J. Geophys. Res., 106, 31,463–31,487.

Trull, T. W., Jansen, P., Schulz, E., Weeding, B., Davies, D. M., and Bray, S. G. (2019). Autonomous multi-trophic observations of productivity and export at the Australian Southern Ocean Time Series (SOTS) reveal sequential mechanisms of physical-biological coupling. Font. Mar. Sci. 6. doi: 10.3389/fmars.2019.00525

Line 220: spell out "foraminifera", rather than "forams" as for the other groups of organisms.
Corrected

Line 231: add semi-colon (";") between "... SOTS site" and "however", rather than a comma (",").
Corrected

Lines 232-234: isn't the apparent non-seasonality in silicoflagellate abundance (Fig. 13) also warranted some discussion here?
Thank you for the comment; it is interesting to note that the silicoflagellates do not mimic the seasonality of the main silica-dependent group, the diatoms. We note that the total abundances of the silicoflagellates sampled from surface waters by the RAS is low, typically 2 orders of magnitude lower than that of the total diatoms. It may simply be that the silicoflagellates are outcompeted for available silica by the diatoms in summer, and that contributes to their low abundances. Additionally, silicoflagellates are only confidently identified during that part of their life-cycle when siliceous exoskeletons are produced – naked stages are poorly resolved and only recently linked to the skeleton-bearing stages (McCartney et al, 2014). Their seasonality is likely more complex than we estimate here. We have expanded section 3.3 in the revised text (lines 250-254)

Lines 249-251: why are no values provided for the flux "transfer efficiency".
Values for transfer efficiency have been provided in the last paragraph of this section (line 279).

Lines 254-257: Long sentence – consider re-writing.
This has been split into two sentences.

Line 261: "... interannual variability" of what specifically?
The citation refers specifically to interannual variability in the $CO_2$ system, this has been added to the revised text (line 303).

Line 261-262: "do you mean "... multiple linear regression analysis and ..." ?
This refers to recent work that expanded the time-series of surface water $pCO_2$ using a multiple linear regression and satellite observations. This has been clarified in the revised text (line 304).

Line 266: add "(NCP" after "Net community production".
Added

Line 272: "The seasonal succession away from a phytoplankton-dominated community" – not sure what this actually means and how this can be "elucidated on the basis of chlorophyll and backscatter data" – please comment.
This has been rephrased to indicate that seasonal succession in the phytoplankton community has been evaluated using chlorophyll fluorescence and optical backscatter (line 314).

Line 273: "ship board" is written as "shipboard" (preferred) and "ship-board" in other places in the text. Similarly, "time scales" (e.g., line 324). Please ensure consistency.
Corrected

Lines 276-275: no PAR data are shown in the paper yet Fe and Mn are inferred as seasonally co-limiting. Is light also a factor, especially in winter with such deep mixed layers?
Thank you for the comment; light also limits phytoplankton growth, and this has been added to the revised text (line 320).

Line 279: "cocolithophores" is missing one of the "c"s.
Corrected

Line 280: "there is a seasonal depletion" – when?
The silicate depletion occurs in the (austral) summer, between January and April; this has been added to the revised text.

Line 281: "silica" should be "silicate" as it refers to the macronutrient, not the mineral.
Corrected, also throughout the document

Line 281: "similar to the global median" – what is this value? Should be stated here for ease of reference.
Added

Line 283: add "carbon" between "biological" and "pump".
Changed throughout

Line 301: "an independent measure of particle export" – was this at the SOTS site or in the SAZ in general?

This comparison was done for the SOTS site in particular, as well as in a broader region of the SAZ. This has been clarified in the revised text (line 349).

Lines 305-316: the work by Wilks et al. on diatom fluxes and species identifications at SOTS is not references here.

Thank you, these have now been included.

Line 306: not sure what "modern and historical properties of diatoms" means.

Thank you, this has been revised.

Line 310: need space between "huxleyi" and "is".

Corrected

Lines 325-337 "Links to Higher Trophic Levels": how do these studies relate back to the climatology work presented in the paper? As outlined here, it appears that this discussion is just outlining other studies without reflecting on the relationships to the new research that is the core aspect of the paper.

Thank you for the comment. The initial goal was to make links between higher trophic levels and the lower trophic and biogeochemical observations made at the SOTS site. We agree that the section as written falls short of this goal, and have removed it from the revised manuscript.

Line 334: do you mean "particulate organic carbon"? Also, "low surface (chlorophyll or plankton) biomass" later in the same sentence?

Yes, the sentence referred to POC, and low surface biomass; this section has been removed from the revised text (see the response to the comment above).

Lines 335-337: the last sentence seems to be somewhat tacked on but does emphasise the other co-benefits of sediment trap records with respect to plankton species identifications.

Thank you for the comment; we have removed the paragraph from the revised text but agree that there is benefit in plankton taxonomy in sediment trap samples, and continue to maintain an archive in case new opportunities for this work should arise.

Line354: "... future impacts." – of what? On what?

This sentence has been clarified to indicate that ongoing observations are required for robust models than can predict climate variability and its impacts on oceanic processes and ecosystems (line 427).

Line 364: "... RV Investigator" should be italicised.

Corrected

Line 420: species name should be italicised.
Corrected, also for other references with species names
Line 425: missing the opening speech bracket from the title.
Corrected

Line 489: extra comma after "... A. S.,". Similarly on line 493.
Corrected

Lines 490, 498 and 501: species names should be italicised.
Corrected

Lines 513-517: do all the authors need to be listed here (Sathyendranath et al.)? Maybe list first 10 and then add "et al." Similarly, lines 549-552 (Bates et al.) and lines 554-557 (Takahashi et al.).
Corrected in line with the journal specifications for referencing.

Figure 1: could the mean circulation patterns and main surface water masses be also shown on this figure, especially for international readers not familiar with this region?
Thank you for the suggestion; we have revised Figure 1 to include (schematic) surface circulation patterns. The surface water masses are difficult to capture in the same figure, and are included in the T-S plot in Fig. 3

Figure 2: by showing the climatologies of the various fronts on this figure raises questions about how variable the SAF and PF are? In particular, do shifts in the location of the SAF affect the SOTS site?
Please see response above; the location of the SAF and PF are less variable than the STF and because the SOTS site is located in the northern part of the SAZ, it is less influenced by their motions.

Figure 3: does "sensor" mean shipboard "CTD" or moored "CTD" data or something else? Are these data from the SOTS site only rather than from other deployments in the wider SAZ? State this in the caption.
The data used to generate Figure 3 are from both ship-board CTD casts and CTD sensor data at the SOTS site (-46S to -47S, 142-143E). This has been added to the caption in the revised text.

Figures 5b and 6: what is the definition that is used to define the "mixed-layer depth" calculations?
Mixed-layer depth is defined by a temperature difference of 0.3C from the topmost sensor closest to 10m, following Weeding & Trull (2014) and Shadwick et al., 2025; 2023. This has been added to the caption for Figure 5.

Figure 7: is there any way of clearly showing the variability associated with the CTD data?
Thank you for the suggestion; we have added standard deviations to the mean seasonal profiles in Figure 7.

**Figure 8b: Why are there no data from 3900 m shown?**
There was no salinity sensor at 3900m, only a pressure and temperature sensor at 3900m; in response to an earlier comment, this figure has been removed from the revised text.

**Figure 8c: the pressure differences shown here suggest the sensors were moored at slightly different depths. So, did this affect other moored data collected at the same time or were the moorings adjusted to account for the slight depth variations between mooring deployments?**
The pressure differences reflect the slight differences in depths of individual moorings; all sensor data is gridded to a uniform depth as part of post-processing. In response to an earlier comment, this figure has been removed from the revised manuscript.

**Figure 9: Axis labels are too small to read. Consider adding labels to each of the plots for ease of clarity.**
The font sizes have been adjusted.

**Figure 9: are all of these climatologies generated from within the mixed layer or from a specific water depth?**
All climatologies are computed from the surface (~1m depth). This has been added to the figure caption for clarity.

**Figure 10: Axis labels are too small to read. Consider adding labels to each of the plots for ease of clarity. Can r2 values be ascribed to the linear relationships in the Alk v S plot?**
The font sizes have been adjusted, and an r2 value has been added for the linear relationship between alkalinity and salinity in the shallow (surface to 800 m) waters.

**Figure 11: Axis labels are too small to read. Consider adding labels to each of the plots for ease of clarity. What production models do the green and orange plots, respectively, in Figure 11c refer to?**
The font sizes have been adjusted. The production models used for the NPP are given in the caption of the Figure (the Vertically Generated Production model - VGPM and Carbon-based Production Model – CbPM).

**Figure 12: the colour scale on these plots, especially 12b, make it very difficult to see the difference sin phytoplankton community composition as described. Please adjust.**
This will be adjusted in the revised version.

**Table A1: add "°S" and "°E" to the last two table columns, respectively.**
This has been added.

**Figures A1 and A2: nice to see these detailed mooring schematics but the small font sizes make it difficult/impossible to read any of the details.**
Thank you for the comment. When we use these drawings, we print them in full (A4) size but appreciate that the font is too small when printed on a standard size paper. We have

ensured that the resolution of the figures permits zooming so that the fonts are legible and have directed the reader to the online and/or digital version of the figures for this purpose in the revised captions.

Figure A3: What water depths were these T and S climatologies from? In Figure A3b, what are the bottom red dashes – extreme outliers or artefacts?
The water depth for both the T and S is ~4500m (as in Figure 8), with the red symbols indicating outliers, and the red line, mean values. However, in response to an earlier comment, we have removed this figure from the revised manuscript.

Figure A4: what water depth were these SEM samples collected from? Presume they were collected using the RAS, which needs to be specified here, presuming they are from the SOTS site and not elsewhere in the SAZ? What was the effect of the mercuric chloride preservation on the ease of phytoplankton IDs?
The SEM samples are sub-sampled from the individual bags after recovery of the RAS from the SOFS mooring, and are then matched to the LM, and nutrient samples. Please see our earlier response about annual, additional CTD samples collected to monitor taxonomic analysis, along with cross-referencing to material analysed by taxonomic experts on the sediment trap material from the SOTS (also using mercuric chloride). There are very limited options for biological preservatives that are suitable for both the chemical analysis (nutrients, total alkalinity) and microscopic analysis of the phyto- and microzooplankton communities, for the 12-month deployments required.

The figure caption has been modified for clarity.

---

## Author Response (AR2)

Reviewer #1

We thank Dr. Nodder for his positive review; below we have provided (in blue text) a response to each comment.

The review undertaken by the authors has made substantive changes to the manuscript that have improved its clarity and suitability for publication. There are still a few typographical and formatting errors that the editorial team should be able to point out to the authors to deal with.

Thank you for the comment, we have been through the manuscript and corrected typographical errors.

The only minor comment is that the revised Figure 1 doesn't have the following information shown on it as indicated in the caption: "the schematic positions of the Zeehan Current (ZC), extensions of the of the East Australian Current (EAC), and a recirculation west of the Tasman Rise following (Herraiz-Borreguero and Rintoul, 2010)." This figure needs to be redrafted as suggested by the caption to show this important physical oceanographic information.

Thank you for the comment, the revised version of Figure 1 (below) has been included in the updated manuscript.

[Figure]

Reviewer #2

This manuscript presents an integrated and climatological perspective of two decades of multidisciplinary observations collected at the Southern Ocean Time Series (SOTS) site, located in the Subantarctic Zone southwest of Tasmania. The study synthesises a vast array of physical, biogeochemical, and ecological data, including hydrography, carbonate chemistry, net primary production, phytoplankton community composition, and carbon export. The long-term and high-frequency nature of the dataset makes this work especially valuable in the context of climate variability, biogeochemical cycling, and ecosystem dynamics in the Southern Ocean, a region critically under-sampled yet disproportionately important in terms of global carbon and heat uptake.

I compliment the authors for the impressive effort in sustaining such a comprehensive oceanographic time-series programme and for compiling the data in a coherent and accessible form. This manuscript has the potential to serve as a benchmark reference for the Subantarctic region and a valuable resource for future comparative studies.

Overall, the authors have satisfactorily addressed the points raised by the previous reviewers, considerably improving the manuscript in its current version. Nevertheless, in its present form, the manuscript still contains a few minor issues which, in my view, may lead to misinterpretation by the reader. Below, I outline some of these minor comments, along with recommendations for strengthening the manuscript. Thus, the manuscript should definitely be published and will be worthwhile for the oceanographic community.

We thank Dr. Monteiro for the positive and constructive feedback. Below we have provided (in blue text) a response to each comment.

Lines 2, 11, 29 and 34: In the abstract and at various points in the introduction, the authors refer to the export of "particulate carbon", but it is unclear whether this refers to organic carbon, inorganic carbon, or both. This only becomes clear later in the Results and Discussion sections. It may be beneficial to clarify this distinction from the abstract onwards.
This change has been made in the abstract as suggested.

Line 41: The phrase "low current region" is vague. Consider using "region of weak mean flow" or specifying the typical current velocity observed at the site.
This change has been made as suggested.

Line 44: In the sentence "There is a seasonal evolution of biomass accumulation", it is not clear which type of biomass is being referred to. I assume the authors mean "phytoplankton biomass", which should be specified to avoid ambiguity.
Yes, phytoplankton biomass; changed as suggested.

Line 49: I suggest clarifying what depth range is meant by "upper water column" in this context.
This has been added.

Lines 49–52: The citation to Cresswell (2000) should be revised, as it describes the physical characteristics of the Zeehan Current but not nutrient distributions. If nutrient depletion is central to the argument, consider citing relevant nutrient climatologies or other biogeochemical sources. Additionally, the claim that the SAMW is "oxygen-rich" should be revised unless a more appropriate reference is provided; Herraiz-Borreguero & Rintoul (2011) do not discuss oxygen concentrations.

The mention of nutrients in the subtropical water has been removed, and an additional reference of the oxygen-rich SAMW has been added.

Line 49: The names of water masses should be written out capitalised, e.g., "Subtropical Water", "Subantarctic Mode Water", etc.

This has been changed.

Line 54: The sentence "Below the SAMW, is cooler, and more saline Antarctic Intermediate Water…" is awkward, especially for non-native speakers. Consider rephrasing it as: "Below the SAMW lies cooler and more saline Antarctic Intermediate Water (AAIW)…".

Rephrased as suggested.

Line 62: As with "upper water column", I suggest specifying what depth range is meant by the term "subsurface" here.

This has been added.

Line 64: I recommend including the website address for the Australian Ocean Data Network (AODN) portal directly in the manuscript text, as well for other relevant data sources, even if these are already provided in the "Data availability" section.

These addresses have been added to the Methods section as suggested.

Line 72: "Total alkalinity" is written out in full for the first time here and later referred to in full and acronym "Alk". I suggest reviewing the manuscript to ensure consistency in introducing full terms and their acronyms; in some cases, terms appear to be defined more than once.

Thank you, these have been corrected.

Line 77: There is a missing full stop before the sentence beginning with "The subsurface instruments…".

Corrected.

Line 105: Although the authors cite several references regarding quality control procedures, it would be useful to include a sentence indicating the average magnitude or typical range of uncertainty for the measured parameters.

Thank you for the suggestion. The magnitude of the uncertainty for temperature, salinity, dissolved oxygen, and inorganic nutrients have been added.

Line 145: Consider rephrasing the sentence "The SOTS site is characterised by deep mixing in the autumn and winter seasons, in some years to depths of roughly 500 m (Fig.

5b), driven by a combination of local heat fluxes (Schulz et al., 2012), and northern Ekman transport of colder waters (Rintoul and England, 2002)." Suggested revision: "The SOTS site is characterised by deep mixing during the autumn and winter seasons, reaching depths of roughly 500 m in some years (Fig. 5b), driven by a combination of local heat fluxes (Schulz et al., 2012) and northward Ekman transport of colder waters (Rintoul and England, 2002)."
Rephrased as suggested.

Line 156: The sentence "Depending on the longitudinal range considered, the STF may be observed as far south as 48°S" could be clarified. Suggested revision: "In the longitudinal band 140–144°E, the STF may be observed as far south as 48°S."
Rephrased as suggested.

Figure 5 caption: It would be helpful to include a description of the "thick grey line" when referring to "130–150°E (thick line)".
This has been added.

Line 160: In the phrase "advection of waters with more polar characteristics", please specify which water masses are being referred to.
This has been clarified as suggested.

Lines 162–164: The meaning of "mean magnitude of the seasonal temperature cycle at the surface is 4°C" is unclear. If referring to the seasonal amplitude (i.e., maximum minus minimum), I suggest rephrasing to make that explicit.
This has been rephrased as suggested.

Lines 162–164: In addition, the statement "with summer maximum of roughly 12°C" appears to be inconsistent with the earlier declaration "temperatures ranging from summer maxima of 15°C to winter minima of ~8°C". Please consider clarifying whether these refer to different time periods, depths, or datasets.
The 12 degree maximum refers to the (smoothed) climatological seasonal cycle, while the 15 degree maximum to the monthly observations; this has been clarified.

Line 176: It is unclear why the sentence begins with "The seasonality in temperature is moderate as described above", as both this sentence and the entire paragraph refer to the carbonate system. Perhaps a more suitable phrasing would be: "Since temperature seasonality is moderate, as described above, the seasonal cycle of $pCO_2$ is dominated by the impact of biological productivity..."
This has been rephrased as suggested.

Line 177: The term "inorganic carbon ($TCO_2$)" has already been defined earlier in the manuscript. Please revise the use of full terms and respective acronyms throughout the manuscript to avoid redundancy.
Corrected.

Line 180: The sentence "The seasonality in alkalinity (Alk) is computed from salinity…" could be more clearly phrased as: "Alk is computed from salinity and thus exhibits similarly modest seasonality".
Rephrased as suggested.

Line 183: The phrase "TCO2-rich water from below" might be clearer if reworded to: "…TCO2-rich water from deeper layers".
Rephrased as suggested.

Lines 195–196: The phrase "Alk ranges from roughly 2280 µmol kg–1 at 500 m to greater than 2350 µmol kg–1 below 1500 m" seems to refer to measured Alk values, rather than those estimated from salinity. If so, it would be helpful to state explicitly that these are "measured Alk". The same applies to the sentence beginning "The relationship between 'measured Alk' and salinity…".
Yes, the quoted ranges are for the measured Alk and not those estimated from salinity. This has been rephrased for clarity.

Line 199: The terms "Subtropical surface waters" and "mode and intermediate waters" should be capitalised as they refer to water masses, i.e., "Subtropical Surface Waters", "Mode Waters", and "Intermediate Waters".
Corrected.

Line 203: The statement "…the SOTS site exhibits the high-nutrient…" is not evidently supported by Figure 2. Consider either providing a different figure or referencing supporting data to substantiate this claim.
Additional references have been added.

Line 261 / Table 2: Under the section "Emergence of Trends", it would be useful to indicate the exact period over which each trend was observed and calculated. This would help readers better contextualise the findings.
This has been added to the caption of Table 2 as suggested.

Lines 314 and 318: The term "calcium carbonate" was defined previously in the manuscript. Please avoid redefining terms unless necessary for clarity.
Corrected.

Line 362: The phrase "Time-series of upper ocean nutrients and biomass…", does it refer specifically to phytoplankton biomass?
Yes, phytoplankton biomass. This has been added to the revised text.

Line 367: "Net community production (NCP)" has already been introduced earlier.
Corrected.

Lines 384–385: I suggest including website address for the OceanSITES network and appropriately referencing GLODAP and SOCAT datasets.
Added.